# STream3R: Scalable Sequential 3D Reconstruction with Causal Transformer

**Yushi Lan**[1*], **Yihang Luo**[1*], **Fangzhou Hong**[1], **Shangchen Zhou**[1],
**Honghua Chen**[1], **Zhaoyang Lyu**[2], **Shuai Yang**[3], **Bo Dai**[4], **Chen Change Loy**[1], **Xingang Pan**[1]
[1]S-Lab, Nanyang Technological University, Singapore
[2]Shanghai Artificial Intelligence Laboratory [3]WICT, Peking University [4]The University of Hong Kong
https://nirvanalan.github.io/projects/stream3r

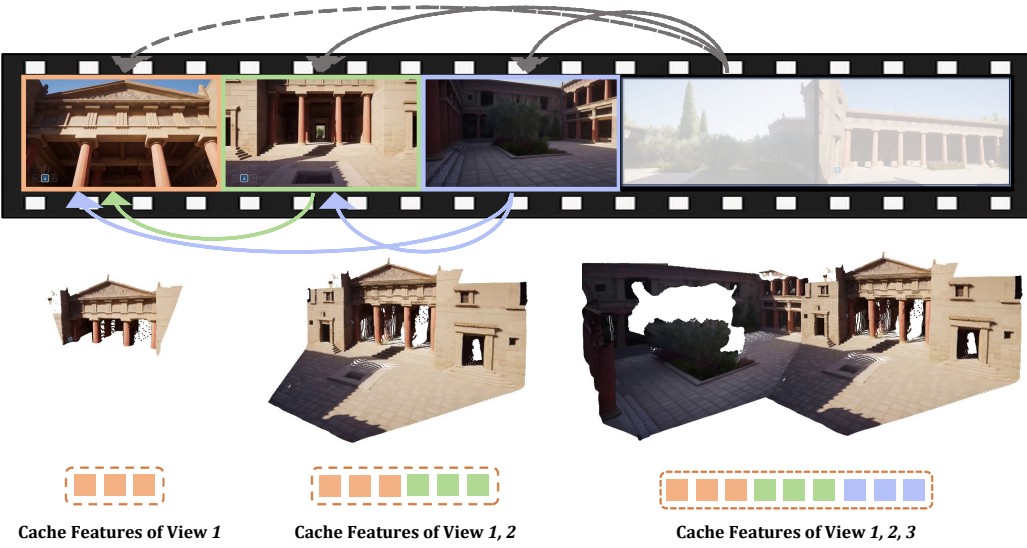

Figure 1: STream3R. Given a stream of input images, our method estimates dense 3D geometry for each incoming frame using a causal Transformer. Features from previously observed frames are cached as context for future inference. The demo video is from Genie 3 (Ball et al., 2025).

## Abstract

We present STream3R, a novel approach to 3D reconstruction that reformulates pointmap prediction as a decoder-only Transformer problem. Existing state-of-the-art methods for multi-view reconstruction either depend on expensive global optimization or rely on simplistic memory mechanisms, both of which scale poorly with sequence length. In contrast, STream3R introduces a streaming framework that efficiently processes image sequences using causal attention, inspired by advances in modern language modeling. By learning geometric priors from large-scale 3D datasets, STream3R generalizes well to diverse and challenging scenarios, including dynamic scenes where traditional methods often fail. Extensive experiments show that our method consistently outperforms prior work across both static and dynamic scene benchmarks. Moreover, STream3R is inherently compatible with LLM-style training infrastructure, enabling efficient large-scale pretraining and fine-tuning for various downstream 3D tasks. Our results highlight the potential of causal Transformer models for online 3D perception, paving the way for real-time 3D understanding in streaming environments.

## 1 Introduction

Reconstructing detailed 3D geometry from images is the crux in computer vision (Schonberger & Frahm, 2016; Schönberger et al., 2016; Chen et al., 2021) and serves as the prerequisite for a series of downstream applications, such as autonomous driving (Geiger et al., 2013), virtual reality (Zheng

et al., 2023; Lan et al., 2024), robotics (Irshad et al., 2024), and more. While traditional visual-geometry methods like SfM (Schonberger & Frahm, 2016) and Multi-view Stereo (Yao et al., 2018; 2019) tackle this problem by solving a series of sub-problems through handcrafted designs, a recent trend led by DUSt3R (Wang et al., 2024d) has demonstrated a promising new way of directly regressing point clouds using powerful transformers. This paradigm, along with its follow-up works including MASt3R (Leroy et al., 2024), Fast3R (Yang et al., 2025), and VGG-T (Wang et al., 2025a), enables the reconstruction of 3D geometry from a number of input images – ranging from a single image to hundreds – offering a more unified solution to 3D reconstruction.

While these works focus on processing a fixed set of images, real-world applications often require continuously processing streaming visual input and updating the reconstruction on-the-fly (Davison et al., 2007), such as when an autonomous agent explores a new environment, or when processing a long video sequence. Handling streaming input poses significant new challenges. For example, naively running Fast3R or VGG-T every time a new image arrives would incur significant redundant computation, as they have to reconstruct from scratch without inheriting previous results. These methods also struggle with long videos due to the expensive full-attention operation. Spann3R (Wang & Agapito, 2024) extends DUSt3R with a memory design (Cheng & Schwing, 2022) to support incremental reconstruction, but it still suffers from significant accumulated drift and fails over dynamic scenes. The most relevant concurrent work is CUT3R (Wang et al., 2025b), which proposes a RNN paradigm (Zaremba et al., 2015) to handle unstructured or streaming inputs. However, the RNN-based design does not scale well with modern network architectures (Dao, 2024) and struggles with long-range dependency due to its limited memory size.

In light of the streaming nature of this task, in this work, we are interested in investigating *the use of a transformer with uni-directional causal attention to achieve online, incremental 3D reconstruction.* In an LLM-style transformer with causal attention, the prediction at each step reuses previous computations through a KVCache, which has been proven successful in many language and audio tasks (Touvron et al., 2023; Copet et al., 2023). We observe that this property is also highly desirable for addressing online 3D reconstruction from streaming data, as each step should build upon the previous reconstruction while integrating new content from the incoming frame.

Motivated by this, we propose STREAM3R, a framework that performs 3D reconstruction from unstructured or streaming input images, and predicts the corresponding point maps in both world and local coordinates (Yang et al., 2025). Unlike concurrent works (Yang et al., 2025; Wang et al., 2025a) that resolve this issue by replacing DUSt3R's asymmetric decoders with bi-directional attention blocks (Devlin et al., 2019; Brooks et al., 2024), STREAM3R follows the modern *decoder-only* (Brown et al., 2020) transformer design, where incoming frames are sequentially processed and registered with causal attention (Chen et al., 2025). In this way, STREAM3R is naturally compatible with modern Large Language Models (LLMs) (Touvron et al., 2023) training and inference techniques such as window attention (Jiang et al., 2023) and KVCache (Brown et al., 2020), i.e., the tokens of processed observations will be saved as reference for registering incoming frames.

We train our method end-to-end on a large collection of 3D data, and benchmark the proposed method on a series of downstream applications. In summary, our key contributions are as follows: (1) We propose STREAM3R, a decoder-only transformer framework that reformulates dense 3D reconstruction as a sequential registration problem with causal attention, enabling scalable processing of unstructured and streaming inputs. (2) The design is naturally compatible with modern LLM-style training and inference paradigms, allowing efficient context accumulation across frames. (3) Our architecture supports both world- and local-coordinate pointmap prediction and extends seamlessly to large-scale novel view synthesis through splatting-based rendering; trained end-to-end on diverse 3D data, STREAM3R achieves competitive or superior performance on standard benchmarks with strong generalization and fast inference.

## 2 RELATED WORK

**Classic 3D Reconstruction.** Early 3D reconstruction pipelines – such as Structure-from-Motion (SfM) (Hartley & Zisserman, 2003; Schonberger & Frahm, 2016; Tang & Tan, 2018) and SLAM (Davison et al., 2007; Mur-Artal et al., 2015; Teed & Deng, 2021) – estimate sparse geometry and camera poses from image collections via geometric reasoning. More recent approaches such as NeRF (Mildenhall et al., 2020; Zhang et al., 2020; Wang et al., 2021a) and Gaussian Splat-

ting (Kerbl et al., 2023; Huang et al., 2024) shift the focus to high-fidelity novel view synthesis using continuous volumetric representations. However, these methods are typically trained per-scene with no learned priors, leading to slow convergence and poor generalization to sparse or occluded inputs – a limitation sometimes referred to as the *tabula rasa* assumption (Wang et al., 2025b). In contrast, we adopt a data-driven approach that learns geometric priors from large-scale 3D datasets (Ling et al., 2024; Reizenstein et al., 2021), enabling fast and generalizable reconstruction from unstructured or streaming inputs.

**Learning 3D Priors from Data.** Recent works leverage large-scale data to learn priors for depth estimation (Yang et al., 2024b; Ke et al., 2024; Hu et al., 2025), pose+depth estimation (Li et al., 2024; Wang et al., 2024b), and bundle adjustment (Wang et al., 2024a). While these methods improve generalization, most focus on monocular depth or two-view setups, limiting their ability to reconstruct full geometry in the absence of known intrinsics (Yin et al., 2023). VGGSfM (Wang et al., 2024a) introduces differentiable bundle adjustment by integrating neural feature matching with classic optimization, but remains iterative and computationally heavy, impeding scalability. In the multi-view stereo domain, approaches such as MVSNeRF (Chen et al., 2021; 2024) and MVSNet (Yao et al., 2018) integrate neural networks into the MVS pipeline but typically require known camera poses and still heavily rely on hand-crafted components to effectively incorporate 3D geometry.

**Pointmap-based Representations.** Pointmap-based representations (Wang et al., 2024d; Leroy et al., 2024; Charatan et al., 2024; Xu et al., 2024; Szymanowicz et al., 2023; Zhang et al., 2024a;b; Fang et al., 2026) have recently emerged as a unifying format for dense 3D geometry prediction, aligning well with the output structure of neural networks. Compared to voxels (Sitzmann et al., 2019), meshes (Gkioxari et al., 2019), or implicit fields (Park et al., 2019; Mildenhall et al., 2020), pointmaps enable feedforward inference and real-time rendering, and can directly support applications such as rasterization-based rendering (Kerbl et al., 2023), SLAM (Murai et al., 2024; Liu et al., 2024), and few-shot synthesis (Ye et al., 2025). DUSt3R (Wang et al., 2024d) and follow-ups like MASt3R (Leroy et al., 2024) recast stereo 3D reconstruction as dense pointmap regression, jointly estimating depth, pose, and intrinsics from image pairs. However, their pairwise design fundamentally limits scalability – requiring quadratic fusion operations and complex global alignment procedures when handling multi-view scenarios. Our approach maintains the advantages of pointmap representations while overcoming these scalability limitations.

**4D Reconstruction from Monocular Videos.** Reconstructing dense geometry of dynamic scenes from monocular video is significant but challenging for conventional methods. Recent methods (Lei et al., 2024; Chu et al., 2024; Li et al., 2024; Kopf et al., 2021) leverage depth priors to resolve this challenge. Specifically, Robust-CVD (Kopf et al., 2021) and MegaSAM (Li et al., 2024) require time-consuming per-video optimization. MonST3R (Zhang et al., 2024a) builds on DUSt3R to output pointmaps for dynamic scenes by fine-tuning DUSt3R on the dynamic datasets. However, it still requires a sliding-window based per-video global alignment as post-processing. In contrast, our method enables feedforward 4D reconstruction directly from monocular videos, supporting online prediction without costly per-video optimization or post-processing alignment.

**Reconstruction Methods from Streaming Inputs.** Streaming approaches offer a more scalable alternative solution for the 3D reconstruction problem, represented by the monocular SLAM pipelines (Davison et al., 2007; Liu et al., 2024; Zhu et al., 2024). Inspired by the existing learning-based online 3D reconstruction methods (Choy et al., 2016; Yu et al., 2021; Wang et al., 2021c), recently Spann3R (Wang & Agapito, 2024) introduces a memory-based extension to DUSt3R, while Fast3R (Yang et al., 2025) and VGG-T (Wang et al., 2025a) replace asymmetric decoders with Transformer-based attention stacks to directly enable multi-view fusion. Despite these advances, these approaches still predominantly rely on global full-attention mechanisms, limiting their real-time scalability with increasing sequence length. CUT3R (Wang et al., 2025b) adopts an RNN-style architecture to process unstructured inputs incrementally, but suffers from limited memory capacity and poor compatibility with modern hardware acceleration techniques (Dao, 2024). Our method fundamentally re-conceptualizes pointmap prediction as a decoder-only Transformer task, enabling efficient causal inference through techniques like KVCache and windowed attention (Jiang et al., 2023; Brown et al., 2020). This architectural design allows us to scale effectively to long sequences while maintaining full compatibility with modern LLM-style training infrastructure and optimization techniques, overcoming the limitations of previous approaches.

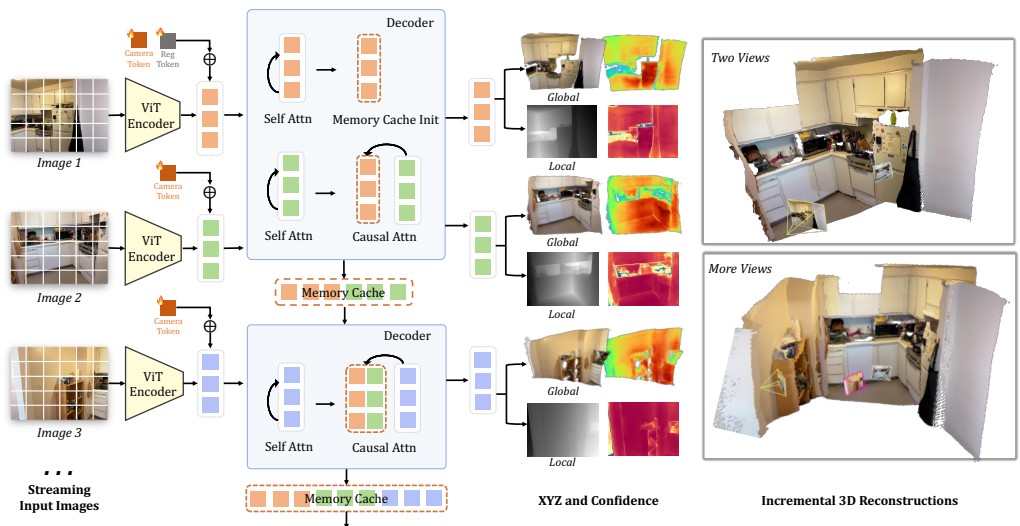

Figure 2: Method overview. Built on a causal transformer, STREAM3R processes streaming images sequentially for 3D reconstruction. Each input image is first tokenized using a shared-weight ViT encoder, and the resulting tokens are passed to our causal decoder. Each decoder layer begins with frame-wise self-attention. For subsequent views, the model applies causal attention to the memory tokens cached from previous observations. The outputs include point maps and confidence maps in both world and camera coordinate systems, as long as the camera pose as shown on the right. Note that we visualize the point cloud of the $\text{Head}_{\text{local}}$ with its depth map.

## 3 PRELIMINARIES: DUSt3R

We reformulate DUSt3R (Wang et al., 2024d) to process streaming images. In DUSt3R, each incoming image $\boldsymbol{I}_t$ is patchified into $K$ tokens $\boldsymbol{F}_t = \text{Encoder}(\boldsymbol{I}_t)$ with $\boldsymbol{F}_t \in \mathbb{R}^{K \times C}$ using a weight-sharing ViT encoder (Dosovitskiy et al., 2021). The model operates on two views at a time ($t \in \{1, 2\}$), producing token sets $\boldsymbol{F}_1$ and $\boldsymbol{F}_2$. Two symmetric decoder branches then perform cross-attentive reasoning through $B$ transformer blocks, defined recursively as $G_t^i = \text{DecoderBlock}_t^i(G_t^{i-1}, G_{3-t}^{i-1})$ for $i = 1, \ldots, B$, with initialization $G_t^0 := \boldsymbol{F}_t$. Finally, each branch predicts a pointmap and confidence map $(\hat{\boldsymbol{X}}_{t,1}, \hat{\boldsymbol{C}}_{t,1}) = \text{Head}_t(G_t^0, \ldots, G_t^B)$. DUSt3R is inherently limited to two-view inputs and relies on an expensive global alignment procedure to incorporate additional views.

## 4 METHOD

We introduce STREAM3R, a transformer that ingests uncalibrated streaming images as inputs and yields a series of 3D attributes as output. The input can be either unstructured image collections or video. Unlike existing approaches (Wang et al., 2025a; Yang et al., 2025) that address this issue by adopting costly bi-directional attention over the entire input sequence or using fixed-size memory buffers (Wang & Agapito, 2024; Wang et al., 2025b), STREAM3R instead caches features from the past frames as *context* and processes incoming frames sequentially using causal attention over the accumulated observations. This design not only enables faster training and quicker convergence but also aligns with the architectural principles of modern LLMs, allowing us to leverage the advances of that domain. We first introduce the problem formulation in Sec. 4.1, the architecture in Sec. 4.2, and the training objectives in Sec. 4.3, and the implementation details in Sec. 5. An overview of the proposed method is shown in Fig. 2. Also note that STREAM3R shares the same architecture design with DUSt3R, and please refer to the appendix for the preliminaries.

### 4.1 PROBLEM DEFINITION AND NOTATION

STREAM3R is a regression model that sequentially takes a stream of $N$ RGB images $(\boldsymbol{I})_t^N$, where each image $\boldsymbol{I} \in \mathbb{R}^{3 \times H \times W}$ belongs to the same 3D scene. The streaming inputs are successively

transformed into a set of 3D annotations corresponding to each frame:

$$f_\theta((\boldsymbol{I})_t^N) = (\hat{\boldsymbol{X}}_t^{\text{local}}, \hat{\boldsymbol{X}}_t^{\text{global}}, \hat{\boldsymbol{P}}_t)_t^N. \tag{1}$$

Technically, STREAM3R is implemented as a causal transformer that maps each image $\boldsymbol{I}_t$ into its corresponding pointmap of the local coordinate $\hat{\boldsymbol{X}}_t^{\text{local}} \in \mathbb{R}^{3 \times H \times W}$ and its pointmap in a global coordinate $\hat{\boldsymbol{X}}_t^{\text{global}} \in \mathbb{R}^{3 \times H \times W}$, which is defined with respect to the camera coordinate of the first input frame $\boldsymbol{I}_1$, and its relative camera pose $\hat{\boldsymbol{P}}_t \in \mathbb{R}^9$ including both intrinsics and extrinsics. We devise later how these 3D attributes are predicted.

## 4.2 Causal Transformer for 3D Regression

**Causal Attention for Long-context 3D Reasoning.** As mentioned in Sec. 3, given the streaming inputs, for each current image, $\boldsymbol{I}_t$, our method first tokenizes it into the features $\boldsymbol{F}_t = \text{Encoder}(\boldsymbol{I}_t)$. The main difference lies in the decoder side: rather than performing bi-directional attention over the whole sequence (Yang et al., 2025) or interacting with a learnable *state* as in Wang et al. (2025b), we draw inspiration from the LLMs (Touvron et al., 2023; Brown et al., 2020; DeepSeek-AI et al., 2024) and perform causal attention efficiently with previous observations. Specifically, after performing frame-wise self-attention in each decoder block, the current feature $G_t^{i-1}$ will cross-attend to the features of previously observed frames corresponding to the same layer:

$$G_t^i = \text{DecoderBlock}^i\left(G_t^{i-1}, \ G_0^{i-1} \oplus G_1^{i-1} \oplus \cdots \oplus G_{t-1}^{i-1}\right). \tag{2}$$

This interaction ensures efficient information transfer to handle long-context dependencies. Note that this operation is easy to implement and well optimized with KV cache during inference for efficient computation (Brown et al., 2020; Touvron et al., 2023).

**Simplified Decoder Design.** To achieve this, several network architecture modifications are required. In DUSt3R, the decoder follows a symmetric design, i.e., two separate decoders $\text{Decoder}_1$, $\text{Decoder}_2$ are employed to handle two input views. To extend to an arbitrary number of inputs, we remove the symmetric design and only retain a *single* decoder $\text{Decoder}$ to process all the input frames. Specifically, each block in the decoder contains a $\text{SelfAttn}$ block for *frame-wise* attention, and a $\text{CrossAttn}$ block for causally attending to the features of all previous observations. Note that we process the first two frames following the convention of DUSt3R due to the lack of historical context. All incoming frames afterwards follow the causal operation in Eq. (2). Note that to indicate the canonical world space, we add a learnable register token [reg] to the tokens of the first frame $\boldsymbol{F}_1 = \boldsymbol{F}_1 + [\text{reg}]$, in an element-wise manner, as shown in Fig. 2. In this way, the model learns to output the global points without introducing $N$ separate decoders. Unlike Yang et al. (2025), we did not impose positional embedding for other frames for simplicity.

**Prediction Heads.** After the decoding operation, the 3D attributes corresponding to each frame can be predicted accordingly. Following existing works (Wang et al., 2025b;a), we predict two sets of point maps $\hat{\boldsymbol{X}}_t^{\text{local}}, \hat{\boldsymbol{X}}_t^{\text{global}}$ with their corresponding confidence maps $\hat{\boldsymbol{C}}_t^{\text{local}}, \hat{\boldsymbol{C}}_t^{\text{global}}$. Specifically, the local point map $\hat{\boldsymbol{X}}_t^{\text{local}}$ is defined in the coordinate frame of the viewing camera, and the global point map $\hat{\boldsymbol{X}}_t^{\text{global}}$ is in the coordinate frame of the first image $\boldsymbol{I}_1$. We use two DPT (Ranftl et al., 2021) heads for point map prediction:

$$\hat{\boldsymbol{X}}_t^{\text{local}}, \hat{\boldsymbol{C}}_t^{\text{local}} = \text{Head}_{\text{local}}(G_t^0, \ldots, G_t^B), \tag{3}$$

$$\hat{\boldsymbol{X}}_t^{\text{global}}, \hat{\boldsymbol{C}}_t^{\text{global}} = \text{Head}_{\text{global}}(G_t^0, \ldots, G_t^B), \tag{4}$$

$$\hat{\boldsymbol{P}}_t = \text{Head}_{\text{pose}}(G_t^0, \ldots, G_t^B), \tag{5}$$

where this redundant prediction has been demonstrated to simplify training (Jiang et al., 2025) and facilitates training on 3D datasets with partial annotations (Liu et al., 2022; Yu et al., 2023).

## 4.3 Training Objective

STREAM3R is trained using a generalized form of the pointmap loss introduced in DUSt3R. Given a sequence of $N$ randomly sampled images, sourced either from a video or an image collection, we train the model to produce pointmap predictions denoted by $\mathcal{X} = \{\hat{\mathcal{X}}^{\text{local}}, \hat{\mathcal{X}}^{\text{global}}\}$, where $\hat{\mathcal{X}}^{\text{local}} = \{\hat{\boldsymbol{X}}_t^{\text{local}}\}_{t=1}^N$ and $\hat{\mathcal{X}}^{\text{global}} = \{\hat{\boldsymbol{X}}_t^{\text{global}}\}_{t=1}^N$. The corresponding confidence scores are denoted as $\hat{\mathcal{C}}$.

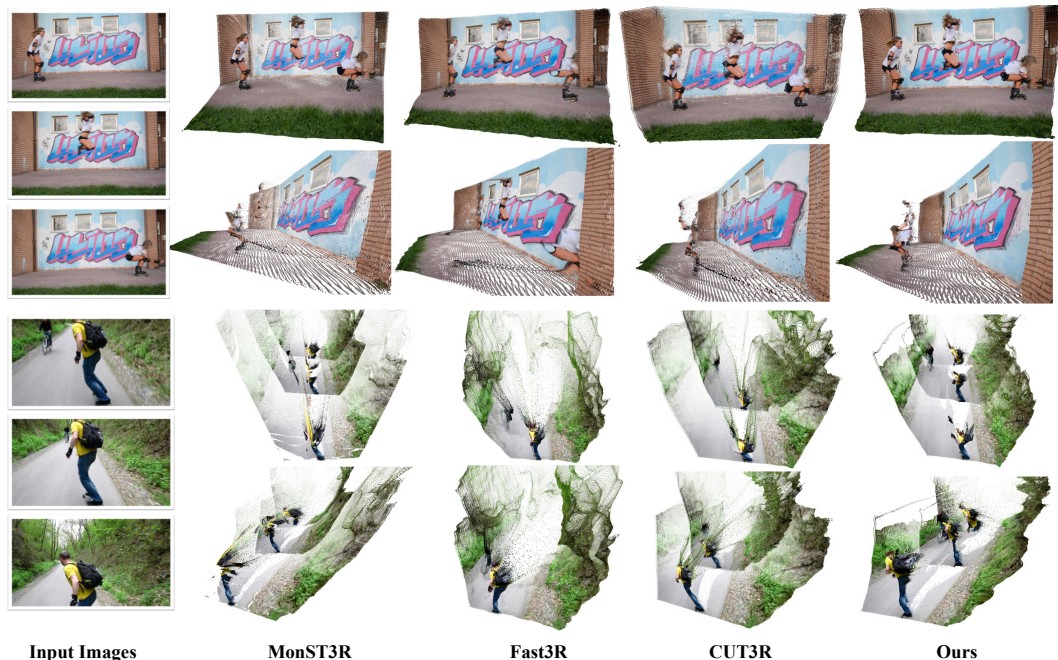

| Input Images | MonST3R | Fast3R | CUT3R | Ours |

Figure 3: Qualitative results on in-the-wild images. We compare our method, $\text{STREAM3R}^{\alpha}$, with MonST3R, Fast3R, and CUT3R, and demonstrate that it achieves superior visual quality.

Following Wang et al. (2025a), we apply a confidence-aware regression loss to the pointmaps: $\mathcal{L}_{\text{conf}} = \sum_{(\hat{\boldsymbol{x}},\hat{c})\in(\hat{\mathcal{X}},\hat{\mathcal{C}})} \left( \hat{c} \cdot \left\| \frac{\hat{\boldsymbol{x}}}{\hat{s}} - \frac{\boldsymbol{x}}{s} \right\|_2 - \alpha \log \hat{c} \right)$, where $\hat{s}$ and $s$ are scale normalization factors for $\hat{\mathcal{X}}$ and $\mathcal{X}$ for scale-invariant supervision (Wang et al., 2024c). We also set $\hat{s} := s$ for metric-scale datasets as in MASt3R (Leroy et al., 2024) to enable metric-scale pointmaps prediction. For the camera prediction loss, we parameterize pose $\hat{\boldsymbol{P}}_t$ as quaternion $\hat{\boldsymbol{q}}_t$, translation $\hat{\boldsymbol{\tau}}_t$ and focal $\hat{\boldsymbol{f}}_t$, and minimize the L2 norm between the prediction and ground truth: $\mathcal{L}_{\text{pose}} = \sum_{t=1}^{N} \left( \|\hat{\boldsymbol{q}}_t - \boldsymbol{q}_t\|_2 + \left\| \frac{\hat{\boldsymbol{\tau}}_t}{\hat{s}} - \frac{\boldsymbol{\tau}_t}{s} \right\|_2 + \left\| \hat{\boldsymbol{f}}_t - \boldsymbol{f}_t \right\|_2 \right)$.

## 5 EXPERIMENTS

**Datasets.** We train our method on a large and diverse collection of 3D datasets, e.g., Co3Dv2 (Reizenstein et al., 2021), ScanNet++ (Yeshwanth et al., 2023), ScanNet (Dai et al., 2017), HyperSim (Roberts et al., 2021), Dynamic Replica (Karaev et al., 2023), DL3DV (Ling et al., 2024), BlendedMVS (Yao et al., 2020), Aria Synthetic Environments (Pan et al., 2023), TartanAir (Wang et al., 2020), MapFree (Arnold et al., 2022), MegaDepth (Li & Snavely, 2018), and ARKitScenes (Baruch et al., 2022). Please check the appendix for the full dataset details.

**Implementation Details.** We provide two versions of STREAM3R, where $\text{STREAM3R}^{\alpha}$ is inspired and fine-tuned from DUSt3R (Wang et al., 2024d) pre-trained weights, and $\text{STREAM3R}^{\beta}$ is initialized from the flagship VGG-T (Wang et al., 2025a) model. For $\text{STREAM3R}^{\alpha}$, we inherit the 24-layer CroCo ViT (Weinzaepfel et al., 2023) as our encoder, and retrofit its 12-layer decoder network by only retaining the first decoder $\text{Decoder} = \text{Decoder}_1$. The DPT-L (Ranftl et al., 2021) heads are used to map the decoded tokens to the local and global point maps accordingly. For $\text{STREAM3R}^{\beta}$, we replace the $\text{SelfAttn}$ layer in the Global Attention of VGG-T with $\text{CausalAttn}$ and fine-tune all the parameters. For memory-efficient and stable training, we inject QK-Norm (Dehghani et al., 2023) to each transformer layer and leverage FlashAttention (Dao, 2024) for BFloat16 mixed precision training.

**Training Details.** Our model is trained with the AdamW optimizer on a batch size of 64 with a learning rate 1e-4 for $400K$ iterations. For each batch, we randomly sample $4-10$ frames from a random training scene. The input frames are cropped into diverse resolutions, ranging from $224 \times 224$ to $512 \times 384$ to improve generalization. The training runs end-to-end on 8 NVIDIA A100 GPUs over seven days. Gradient checkpointing is also adopted to optimize memory usage.

Table 1: Single-frame depth evaluation. We report the performance on Sintel, Bonn, KITTI, and NYU-v2 (static) datasets. The best and second best results in each category are **bold** and underlined respectively. Our method achieves better or comparable performance against existing methods.

| Method | Sintel | | Bonn | | KITTI | | NYU-v2 | |
|---|---|---|---|---|---|---|---|---|
| | Abs Rel ↓ | $\delta$<1.25 ↑ | Abs Rel ↓ | $\delta$<1.25 ↑ | Abs Rel ↓ | $\delta$<1.25 ↑ | Abs Rel ↓ | $\delta$<1.25 ↑ |
| VGG-T (Wang et al., 2025a) | **0.271** | **67.7** | **0.053** | **97.3** | 0.076 | 93.3 | **0.060** | **94.8** |
| Fast3R (Yang et al., 2025) | 0.502 | 52.8 | 0.192 | 77.3 | 0.129 | 81.2 | 0.099 | 88.9 |
| DUSt3R (Wang et al., 2024d) | 0.424 | 58.7 | 0.141 | 82.5 | 0.112 | 86.3 | 0.080 | 90.7 |
| MASt3R (Leroy et al., 2024) | 0.340 | 60.4 | 0.142 | 82.0 | 0.079 | **94.7** | 0.129 | 84.9 |
| MonST3R (Zhang et al., 2024a) | 0.358 | 54.8 | 0.076 | 93.9 | 0.100 | 89.3 | 0.102 | 88.0 |
| Spann3R (Wang & Agapito, 2024) | 0.470 | 53.9 | 0.118 | 85.9 | 0.128 | 84.6 | 0.122 | 84.9 |
| CUT3R (Wang et al., 2025b) | 0.428 | 55.4 | 0.063 | 96.2 | 0.092 | 91.3 | 0.086 | 90.9 |
| STREAM3R$^\alpha$ | 0.350 | 59.0 | 0.075 | 93.4 | 0.088 | 91.3 | 0.091 | 89.9 |
| STREAM3R$^\beta$ | **0.228** | **70.7** | **0.061** | **96.7** | **0.063** | **95.5** | **0.057** | **95.7** |

**Baselines.** We compare our methods against a set of baselines that are designed to take a pair of views as input: DUSt3R (Wang et al., 2024d), MASt3R (Leroy et al., 2024), and MonST3R (Zhang et al., 2024a). Besides, we include the comparison against concurrent methods Spann3R (Wang & Agapito, 2024), CUT3R (Wang et al., 2025b), SLAM3R (Liu et al., 2024), and Fast3R (Yang et al., 2025) that are specifically designed for handling a varying number of input images. We also include the flagship 3D geometry model VGG-T (Wang et al., 2025a) for reference. Note that Fast3R and VGG-T are bi-directional attention methods, and we group them together with methods that require global optimization (GA). We group other concurrent methods together as streaming methods that support processing sequential inputs. Note that for all methods except for VGG-T and STREAM3R$^\beta$, we conduct inference with the largest dimension of 512. For VGG-T based methods, we conduct inference with the largest dimension of 518 due to the requirement of DINO-V2 tokenizer (Oquab et al., 2023). Regarding FPS, we benchmark the inference speed on the A100 GPU with FP32. Comparisons of more concurrent methods (Zhuo et al., 2025; Yang et al., 2024b) are included in the appendix.

## 5.1 MONOCULAR AND VIDEO DEPTH ESTIMATION

**Mono-Depth Estimation.** Following previous methods (Zhang et al., 2024a; Wang et al., 2025b), we first evaluate monocular depth estimation on Sintel (Butler et al., 2012), Bonn (Palazzolo et al., 2019), KITTI (Geiger et al., 2013), and NYU-v2 (Silberman et al., 2012) datasets, which cover dynamic and static, indoor and outdoor, realistic and synthetic data. These datasets are not used for training and are suitable for benchmarking the zero-shot performance across different domains. Our evaluation includes the absolute relative error (Abs Rel) and percentage of inlier points within a 1.25-factor of true depth $\delta < 1.25$, following the convention of existing methods (Hu et al., 2025; Yang et al., 2024a). Per-frame median scaling is imposed as in DUSt3R. We include the quantitative results in Tab. 1. As can be seen, our method achieves state-of-the-art compared to streaming-based methods, and even performs best compared to VGG-T on Sintel, KITTI, and NYU-2. Also note that our method uses fewer datasets and compute resources compared to CUT3R. Specifically, CUT3R adopts a curriculum training of four stages for $100 + 35 + 40 + 10 = 185$ epochs, while our method is trained end-to-end for only 7 epochs using a partial of CUT3R's datasets due to the computational resources constraints.

**Video Depth Estimation.** We also benchmark our model on the video depth task, which evaluates both per-frame depth quality and inter-frame depth consistency by aligning the output depth maps to the ground truth depth maps using a given per-sequence scale. Metric point map methods like MASt3R, CUT3R, and ours are also reported without alignment. The quantitative results for both methods are included in Tab. 2. Over per-sequence scale alignment, our method surpasses optimization-based baselines DUSt3R-GA (Wang et al., 2024d) and MASt3R-GA (Leroy et al., 2024) (static-scene assumption) and even MonST3R-GA (Zhang et al., 2024a) (dynamic-scene, optical flow (Teed & Deng, 2020) dependent). Against the streaming state-of-the-art CUT3R, we achieve higher accuracy on all three benchmarks while running $40\%$ faster. STREAM3R also outperforms full-attention Fast3R (Yang et al., 2025), streaming approaches Spann3R (Wang & Agapito, 2024), and the flagship model VGG-T on Sintel. Notably, STREAM3R$^\beta$-W, using sliding-window attention (Jiang et al., 2023) for constant cache, exceeds STREAM3R$^\beta$ on Bonn and KITTI despite accessing only five past frames.

Table 2: Video depth evaluation. We evaluate scale-invariant and metric depth accuracy on the Sintel, Bonn, and KITTI datasets. Methods that require global alignment are denoted as "GA". The "Type" column indicates whether the method is Optimization-based ("Optim"), streaming ("Stream"), or full-attention ("FA"). We also report inference speed in FPS on the KITTI dataset using $512\times144$ resolution for all methods on an A100 GPU, except for Spann3R, which supports $224\times224$ inputs. Our method achieves performance that is better than CUT3R, while offering faster inference. $\text{STREAM3R}^{\beta}$-W[5] uses sliding window attention on $\text{STREAM3R}^{\beta}$ with window size 5. Note that $\text{STREAM3R}^{\beta}$-W[5] achieves the fastest FPS among all streaming-based methods.

| Alignment | Method | Type | Sintel | | Bonn | | KITTI | | FPS |
|---|---|---|---|---|---|---|---|---|---|
| | | | Abs Rel ↓ | $\delta <$ 1.25 ↑ | Abs Rel ↓ | $\delta <$ 1.25 ↑ | Abs Rel ↓ | $\delta <$ 1.25 ↑ | |
| Per-sequence scale | VGG-T (Wang et al., 2025a) | FA | **0.297** | **68.8** | **0.055** | **97.1** | **0.073** | **96.5** | 7.32 |
| | Fast3R (Yang et al., 2025) | FA | 0.653 | 44.9 | 0.193 | 77.5 | 0.140 | 83.4 | **47.23** |
| | DUSt3R-GA (Wang et al., 2024d) | Optim | 0.656 | 45.2 | 0.155 | 83.3 | 0.144 | 81.3 | 0.76 |
| | MASt3R-GA (Leroy et al., 2024) | Optim | 0.641 | 43.9 | 0.252 | 70.1 | 0.183 | 74.5 | 0.31 |
| | MonST3R-GA (Zhang et al., 2024a) | Optim | 0.378 | 55.8 | 0.067 | 96.3 | 0.168 | 74.4 | 0.35 |
| | Spann3R (Wang & Agapito, 2024) | Stream | 0.622 | 42.6 | 0.144 | 81.3 | 0.198 | 73.7 | 13.55 |
| | CUT3R (Wang et al., 2025b) | Stream | 0.421 | 47.9 | 0.078 | 93.7 | 0.118 | 88.1 | 16.58 |
| | $\text{STREAM3R}^{\alpha}$ | Stream | 0.478 | 51.1 | 0.075 | 94.1 | 0.116 | 89.6 | 23.48 |
| | $\text{STREAM3R}^{\beta}$ | Stream | **0.264** | **70.5** | 0.069 | 95.2 | **0.080** | 94.7 | 12.95 |
| | $\text{STREAM3R}^{\beta}$-W[5] | Stream | 0.279 | 68.6 | **0.064** | 96.7 | 0.083 | 95.2 | 32.93 |
| Metric scale | MASt3R-GA (Leroy et al., 2024) | Optim | **1.022** | 14.3 | 0.272 | 70.6 | 0.467 | 15.2 | 0.31 |
| | CUT3R (Wang et al., 2025b) | Stream | 1.029 | **23.8** | 0.103 | 88.5 | **0.122** | **85.5** | 16.58 |
| | $\text{STREAM3R}^{\alpha}$ | Stream | 1.041 | 21.0 | **0.084** | **94.4** | 0.234 | 57.6 | 23.48 |

Table 3: 3D reconstruction evaluation on 7-Scenes (Shotton et al., 2013). Despite operating in the streaming setting, our method delivers competitive performance, matching or even exceeding that of offline optimization-based methods that leverage global alignment.

| Method | Type | Acc↓ | | Comp↓ | | NC↑ | | FPS |
|---|---|---|---|---|---|---|---|---|
| | | Mean | Med. | Mean | Med. | Mean | Med. | |
| VGG-T (Wang et al., 2025a) | FA | **0.087** | **0.039** | **0.091** | **0.039** | **0.787** | **0.890** | 12.00 |
| Fast3R (Yang et al., 2025) | FA | 0.164 | 0.108 | 0.163 | 0.080 | 0.686 | 0.775 | **30.92** |
| DUSt3R-GA (Wang et al., 2024d) | Optim | 0.146 | 0.077 | 0.181 | 0.067 | 0.736 | 0.839 | 0.68 |
| MASt3R-GA (Leroy et al., 2024) | Optim | 0.185 | 0.081 | 0.180 | 0.069 | 0.701 | 0.792 | 0.34 |
| MonST3R-GA (Zhang et al., 2024a) | Optim | 0.248 | 0.185 | 0.266 | 0.167 | 0.672 | 0.759 | 0.39 |
| Spann3R (Wang & Agapito, 2024) | Stream | 0.298 | 0.226 | 0.205 | 0.112 | 0.650 | 0.730 | 12.97 |
| SLAM3R (Liu et al., 2024) | Stream | 0.287 | 0.155 | 0.226 | 0.066 | 0.644 | 0.720 | **38.40** |
| CUT3R (Wang et al., 2025b) | Stream | 0.126 | 0.047 | 0.154 | 0.031 | 0.727 | 0.834 | 17.00 |
| $\text{STREAM3R}^{\alpha}$ | Stream | 0.148 | 0.077 | 0.177 | 0.058 | 0.700 | 0.801 | 26.40 |
| $\text{STREAM3R}^{\beta}$ | Stream | **0.122** | **0.044** | **0.101** | **0.038** | **0.746** | **0.856** | 20.12 |

## 5.2 3D RECONSTRUCTION

We also benchmark scene-level 3D reconstruction on the 7-scenes (Shotton et al., 2013) dataset and use accuracy (Acc), completion (Comp), and normal consistency (NC) metrics, following the convention of existing methods (Wang & Agapito, 2024; Wang et al., 2025b; 2024d). Following CUT3R, we assess the model's performance on image collections with minimal or no overlap by evaluating using sparsely sampled images, i.e., 3 to 5 frames per scene. The quantitative results are included in Tab. 3. Our method achieves better performance compared to optimization-based methods and strong baselines including Spann3R, Fast3R, CUT3R, and SLAM3R. Compared to CUT3R, our method shows better performance with over 50% times faster during the inference. While SLAM3R achieves the fastest inference, it yields noticeably lower reconstruction accuracy than our method. This performance gap can be partially attributed to SLAM3R being trained and evaluated at a lower input resolution of $224 \times 224$. The comparison results on NRGBD (Azinović et al., 2022) benchmark are included in the appendix.

## 5.3 ABLATION ON THE EFFECTIVENESS OF THE PROPOSED ARCHITECTURE

Here, we conduct detailed ablation analysis on STREAM3R to demonstrate the effectiveness of its designs. Due to the extensive computational resources required to train the model, we only train the ablation models on $224 \times 224$ resolution images. All the datasets are included to train the models.

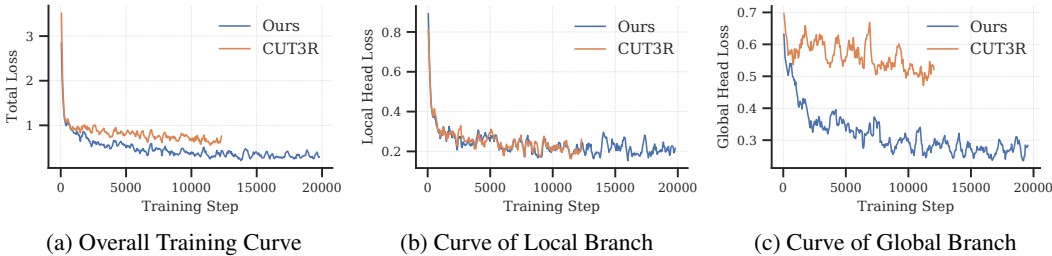

|  (a) Overall Training Curve | (b) Curve of Local Branch | (c) Curve of Global Branch |

Figure 4: Ablation of our proposed STREAM3R. Compared to Wang et al. (2025b), our decoder-only network yields better convergence with faster training speed in the 3D point map prediction task, especially in the global branch.

For a fair comparison, we initialize all the models below using the pre-trained MASt3R (Leroy et al., 2024) checkpoints and train the models using the same hyper-parameters and compute resources.

We demonstrate the effectiveness of decoder-only transformer against RNN design in the sequential 3D pointmap prediction. The main baseline is CUT3R (Wang et al., 2025b), which leverages the RNN design to achieve this. For a fair comparison, we re-train CUT3R and our method using the same dataset and pre-trained model weights initialization. We include the training curve in Fig. 4a, where both models are trained with the same hyperparameters and compute resources. As can be observed, STREAM3R converges faster compared to CUT3R and performs $60\%$ more training steps within the given time. This may sound counterintuitive since STREAM3R is attending to a longer context against CUT3R's constant *state* memory. However, since CUT3R architecture requires a *state-update* operation after each *state-readout* interaction, while STREAM3R directly attends to cached features of existing observations.

We also notice in Fig. 4b that the convergence of $\text{Head}_{\text{local}}$ is similar among the two architectures, while for $\text{Head}_{\text{global}}$, our proposed architecture shows noticeably faster convergence speed, as shown in Fig. 4c. This demonstrates that using a single *state* makes the model harder to register incoming frames due to the limited memory capacity.

Quantitatively, we benchmark the ablation models on both the video depth estimation and 3D reconstruction in Tab. 4, which evaluates the $\text{Head}_{\text{local}}$ and $\text{Head}_{\text{global}}$ correspondingly. For a fair comparison, we evaluate the checkpoints trained for the same number of iterations. As can be observed, our proposed architecture consistently achieves better performance on both tasks.

Table 4: Ablation on video depth estimation and 3D reconstruction. Comparison between RNN-based CUT3R and our proposed architecture STREAM3R$^{\alpha}$. Results show consistent improvements across both video depth estimation (Sintel, BONN, KITTI) and 3D reconstruction (7-Scenes).

| Method | Video Depth Estimation | | | | | | 3D Reconstruction (7-Scenes) | | | | | |
|---|---|---|---|---|---|---|---|---|---|---|---|---|
| | Sintel | | BONN | | KITTI | | Acc↓ | | Comp↓ | | NC↑ | |
| | Abs Rel | $\delta < 1.25$ | Abs Rel | $\delta < 1.25$ | Abs Rel | $\delta < 1.25$ | Mean | Med. | Mean | Med. | Mean | Med. |
| CUT3R | 0.598 | 40.7 | 0.102 | 90.7 | 0.157 | 77.4 | 0.480 | 0.365 | 0.330 | 0.148 | 0.555 | 0.583 |
| STREAM3R$^{\alpha}$ | **0.535** | **47.0** | **0.083** | **94.2** | **0.141** | **81.8** | **0.328** | **0.261** | **0.255** | **0.095** | **0.605** | **0.659** |

## 5.4 MORE ANALYSIS

**Memory Usage.** Tab. 5 illustrates the peak GPU memory usage comparison under different numbers of input frames. All measurements are conducted on a single NVIDIA A100 GPU using FlashAttention (Dao, 2024), with input image resolution set to $448 \times 448$. While naive attention implementations cause quadratic memory usage with respect to sequence length, FlashAttention reduces this from quadratic to linear. Unlike bi-directional methods that process all views jointly, our causal version processes streaming views sequentially, resulting in linearly increasing KV Cache memory.

Our method naturally supports sliding window attention without requiring any fine-tuning. We implement STREAM3R-W[5], a window attention mechanism that always attends to the features of the first frame and the five most recent frames from previous observations. With this approach, the KV Cache size remains constant regardless of input sequence length. As shown in Tab. 2, using window attention achieves comparable or even better performance in video depth evaluation.

**Robustness of the Anchor View.** Using the first frame as the global coordinate system is a standard convention across DUSt3R and its follow-up works, including MASt3R, MonST3R, CUT3R, VGGT, and ours. As shown in the Fig. 7b, even when the first frame has very little overlap, our model still shows strong implicit relative pose-learning capability for the other views.

Quantitatively, we further follow the degradation pipeline of Real-ESRGAN (Wang et al., 2021d) to corrupt the first frame of each sequence, and then evaluate VGGT, CUT3R, and our method on the 7-Scenes dataset. This directly examines scenarios where the first frame is low-quality. As shown in Tab. 10, all methods experience some degradation. However, CUT3R's Accuracy error

Table 5: GPU memory usage comparison (GB).

| Input Frames | 1 | 20 | 40 | 60 | 80 | 100 |
|---|---|---|---|---|---|---|
| VGG-T | 4.70 | 9.99 | 18.66 | 30.48 | 45.47 | 63.63 |
| CUT3R | 3.34 | 3.71 | 4.11 | 4.48 | 4.86 | 5.25 |
| MonST3R-GA | 3.05 | 12.36 | 22.52 | 32.69 | 42.81 | 52.96 |
| STREAM3R$^\alpha$ | 3.02 | 5.64 | 8.31 | 10.98 | 13.65 | 16.32 |
| STREAM3R$^\beta$ | 4.70 | 6.29 | 8.71 | 11.83 | 14.95 | 18.08 |
| STREAM3R$^\alpha$-W[5] | 3.02 | 3.72 | 3.72 | 3.72 | 3.72 | 3.72 |
| STREAM3R$^\beta$-W[5] | 4.70 | 5.18 | 5.18 | 5.18 | 5.18 | 5.18 |

increases markedly from 0.126 to 0.335, whereas that of STREAM3R rises only from 0.122 to 0.223, indicating that our method is considerably more robust under such challenging conditions. We further add visualizations for unordered image inputs and even the case with the non-overlapping anchoring view in Fig. 7. Fig. 7a demonstrates that STREAM3R also performs well on unordered inputs, beyond the streaming setting. Fig. 7b further shows that when the first frame has very little overlap, our model still yields strong implicit relative pose-learning capability for the other views.

## 5.5 MORE APPLICATIONS

**Integration with Novel View Synthesis.** We demonstrate the utility of our method for downstream applications by integrating it with Novel View Synthesis. Specifically, we utilize the dense point maps and camera poses predicted by our model as a geometric prior to initialize 3D Gaussian Splatting (Kerbl et al., 2023). By exporting our predictions to a COLMAP-compatible format (Schonberger & Frahm, 2016), we enable the effective optimization of 3D Gaussians on complex video sequences without relying on external Structure-from-Motion tools. As shown in Fig. 5, STREAM3R-initialized point clouds and camera poses facilitate high-quality novel view renderings.

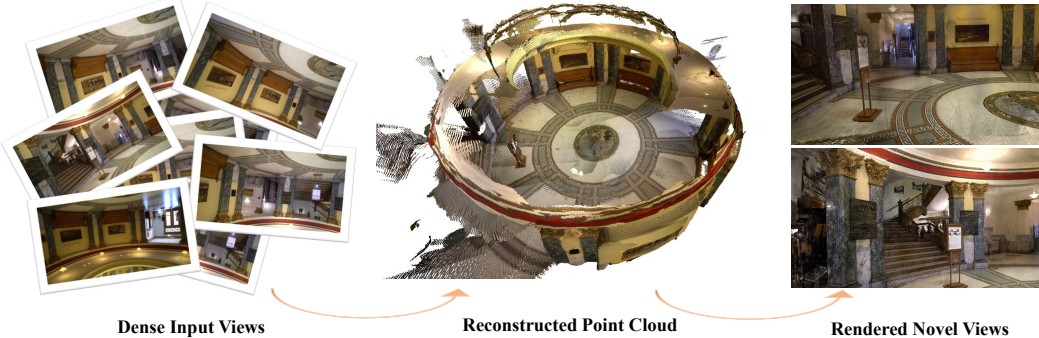

Dense Input Views      Reconstructed Point Cloud      Rendered Novel Views

Figure 5: Visualizations of our reconstructed point cloud and rendered novel view from 3D Gaussian Splatting (Kerbl et al., 2023).

## 6 CONCLUSION

We have introduced STREAM3R, a decoder-only transformer framework for dense 3D reconstruction from unstructured or streaming image inputs. By reformulating reconstruction as a sequential registration task with causal attention, STREAM3R overcomes the scalability bottlenecks of prior work and aligns naturally with LLM-style training and inference pipelines. Our design allows efficient integration of geometric context across frames, supports dual-coordinate pointmap prediction, and generalizes to novel-view synthesis over large-scale scenes without requiring global post-processing. Through extensive experiments across standard benchmarks, we show that STREAM3R achieves competitive or superior performance in the monocular/video-depth estimation and 3D reconstruction tasks, with significantly improved inference efficiency. By bridging geometric learning with scalable sequence modeling, we hope this work paves the way for more general-purpose, real-time 3D understanding systems. Please refer to appendix for the limitation discussion.

**Acknowledgement.** This research is supported by the National Research Foundation, Singapore, under its NRF Fellowship Award NRF-NRFF16-2024-0003. This research is also supported by cash and in-kind funding from NTU S-Lab and industry partner(s).

## 7 REPRODUCIBILITY STATEMENT

We exclusively use publicly available datasets for model training, with complete details provided in the paper. All code and model checkpoints are released here.

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

# A  APPENDIX

## A.1  USE OF LARGE LANGUAGE MODELS

Large Language Models (LLMs) are used exclusively for minor grammar corrections and stylistic polishing of the manuscript. They are not involved in the design of the methodology, execution of experiments, analysis of results, or any other aspect of the scientific contribution.

## A.2  DATASET DETAILS

We train our model on 29 datasets that contains a diverse range of scene types, including static and dynamic scene and objects. Specifically, we mainly follow the data splits of CUT3R (Wang et al., 2025b), and the main 15 datasets with highest sampling ratio are: Co3Dv2 (Reizenstein et al., 2021), ScanNet++ (Yeshwanth et al., 2023), ScanNet (Dai et al., 2017), HyperSim (Roberts et al., 2021), Dynamic Replica (Karaev et al., 2023), DL3DV (Ling et al., 2024), BlendedMVS (Yao et al., 2020), Aria Synthetic Environments (Pan et al., 2023), TartanAir (Wang et al., 2020), MapFree (Arnold et al., 2022), MegaDepth (Li & Snavely, 2018), WildRGBD (Xia et al., 2024), Waymo (Sun et al., 2020), Bedlam (Black et al., 2023), and ARKitScenes (Baruch et al., 2022). We do not include 3D Ken Burns (Niklaus et al., 2019), IRS (Wang et al., 2021b), and SmartPortraits (Kornilova et al., 2022) for training since these datasets are either single view or fail to download successfully. We adapt the official scripts provided by CUT3R (Wang et al., 2025b), DUSt3R (Wang et al., 2024d), and Spann3R (Wang & Agapito, 2024) for dataset processing. For training $\text{STREAM3R}^{\beta}$, we remove all the single-view datasets as in VGG-T, leaving 19 datasets for training. We did not find performance degradation when removing the single-view datasets. Please refer to the Tab. 6 of the CUT3R for more dataset details.

## A.3  MORE IMPLEMENTATION DETAILS

**More Training Details.** Our method conducts end-to-end training on all datasets on a hybrid of 12 different resolutions, ranging from $224 \times 224$ to $512 \times 384$. Data augmentation side, we perform sequence-level color jittering by applying the same color jitter across all frames in a sequence.

**Network Architecture Details.** We follow DUSt3R and use the CroCoNet (Weinzaepfel et al., 2023) pre-trained ViT for the encoder and decoder design. We directly use the DPT (Ranftl et al., 2021) head for $\text{Head}_{\text{global}}$ and $\text{Head}_{\text{local}}$ implementation. We apply RoPE to the query and key feature before each attention operation for the ViT encoder, but ignore it for the ViT decoder to generalize to an arbitrary number of input views. For ablation studies, we train our model on the same datasets but at resolution $224 \times 224$.

For the sliding window attention version $\text{STREAM3R}^{\beta}$-W[5], we always include the tokens of the first frame to keep the canonical coordinate space unchanged. We set window size W= 5 since it trades off performance and speed, and other window size also stably works. For the full attention version $\text{STREAM3R}^{\beta}$-FA, we directly use the causally trained model $\text{STREAM3R}^{\beta}$ and remove the causal mask in the $\text{SelfAttn}$. This is similar to the "revisit" operation in CUT3R.

## A.4  MORE COMPARISONS AND ANALYSIS

**Video Depth Estimation.** We further expand the video depth comparison in the main paper and include a wider range of baseline methods, including single-frame depth methods Marigold (Ke et al., 2024) and DepthAnything-V2 (Yang et al., 2024c), video depth approaches NVDS (Wang et al., 2023), DepthCrafter (Hu et al., 2025), and ChronoDepth (Shao et al., 2024), and recent joint depth-and-pose estimation methods such as Robust-CVD (Bârsan et al., 2018), CausalSAM (Zhang et al., 2022), DUSt3R (Wang et al., 2024d), MASt3R (Leroy et al., 2024), MonST3R (Zhang et al., 2024a), and Spann3R (Wang & Agapito, 2024). Extended results are shown in Tab. 6. $\text{STREAM3R}^{\alpha}$ consistently outperforms its RNN-based counterpart CUT3R under the per-sequence scale & shift setting, and even achieves state-of-the-art performance on the KITTI dataset while also being the fastest in terms of FPS. Moreover, $\text{STREAM3R}^{\beta}$ delivers even stronger results, attaining the best overall accuracy across the per-sequence scale & shift setting.

Table 6: Video depth evaluation. We report scale&shift-invariant depth, scale-invariant depth and metric depth accuracy on Sintel, Bonn, and KITTI datasets. Methods requiring global alignment are marked "GA", while "Optim" and "Stream" indicate Optimzation-based and Streamne methods, respectively. We also report the FPS on KITTI dataset using $512\times 144$ image resolution for all methods, except Spann3R which Stream supports $224\times224$ inputs.

| Alignment | Method | Type | Sintel Abs Rel ↓ | Sintel $\delta<1.25$ ↑ | BONN Abs Rel ↓ | BONN $\delta<1.25$ ↑ | KITTI Abs Rel ↓ | KITTI $\delta<1.25$ ↑ | FPS |
|---|---|---|---|---|---|---|---|---|---|
| Per-sequence scale & shift | Marigold (Ke et al., 2024) | Stream | 0.532 | 51.5 | 0.091 | 93.1 | 0.149 | 79.6 | <0.1 |
| | Depth-Anything-V2 (Yang et al., 2024c) | Stream | 0.367 | 55.4 | 0.106 | 92.1 | 0.140 | 80.4 | 3.13 |
| | NVDS (Wang et al., 2023) | Stream | 0.408 | 48.3 | 0.167 | 76.6 | 0.253 | 58.8 | - |
| | ChronoDepth (Shao et al., 2024) | Stream | 0.687 | 48.6 | 0.100 | 91.1 | 0.167 | 75.9 | 1.89 |
| | DepthCrafter (Hu et al., 2025) | Stream | 0.292 | 69.7 | 0.075 | 97.1 | 0.110 | 88.1 | 0.97 |
| | Robust-CVD (Kopf et al., 2021) | Stream | 0.703 | 47.8 | - | - | - | - | - |
| | CasualSAM (Zhang et al., 2022) | Optim | 0.387 | 54.7 | 0.169 | 73.7 | 0.246 | 62.2 | - |
| | DUSt3R-GA (Wang et al., 2024d) | Optim | 0.531 | 51.2 | 0.156 | 83.1 | 0.135 | 81.8 | 0.76 |
| | MASt3R-GA (Leroy et al., 2024) | Optim | 0.327 | 59.4 | 0.167 | 78.5 | 0.137 | 83.6 | 0.31 |
| | MonST3R-GA (Zhang et al., 2024a) | Optim | 0.333 | 59.0 | 0.066 | 96.4 | 0.157 | 73.8 | 0.35 |
| | Spann3R (Wang & Agapito, 2024) | Stream | 0.508 | 50.8 | 0.157 | 82.1 | 0.207 | 73.0 | 13.55 |
| | CUT3R (Wang et al., 2025b) | Stream | 0.540 | 55.7 | 0.074 | 94.5 | 0.106 | 88.7 | 16.58 |
| | STREAM3R$^\alpha$ | Stream | 0.356 | 58.6 | 0.068 | 95.7 | 0.099 | 91.0 | 23.48 |
| | STREAM3R$^\beta$ | Stream | 0.205 | 70.8 | 0.062 | 97.4 | 0.071 | 95.1 | 12.95 |
| Per-sequence scale | DUSt3R-GA (Wang et al., 2024d) | Optim | 0.656 | 45.2 | 0.155 | 83.3 | 0.144 | 81.3 | 0.76 |
| | MASt3R-GA (Leroy et al., 2024) | Optim | 0.641 | 43.9 | 0.252 | 70.1 | 0.183 | 74.5 | 0.31 |
| | MonST3R-GA (Zhang et al., 2024a) | Optim | 0.378 | 55.8 | 0.067 | 96.3 | 0.168 | 74.4 | 0.35 |
| | Spann3R (Wang & Agapito, 2024) | Stream | 0.622 | 42.6 | 0.144 | 81.3 | 0.198 | 73.7 | 13.55 |
| | Fast3R (Yang et al., 2025) | FA | 0.653 | 44.9 | 0.193 | 77.5 | 0.140 | 83.4 | 47.23 |
| | CUT3R (Wang et al., 2025b) | Stream | 0.421 | 47.9 | 0.078 | 93.7 | 0.118 | 88.1 | 16.58 |
| | STREAM3R$^\alpha$ | Stream | 0.478 | 51.1 | 0.075 | 94.1 | 0.116 | 89.6 | 23.48 |
| | STREAM3R$^\beta$ | Stream | 0.264 | 70.5 | 0.069 | 95.2 | 0.080 | 94.7 | 12.95 |
| Metric scale | MASt3R-GA (Leroy et al., 2024) | Optim | 1.022 | 14.3 | 0.272 | 70.6 | 0.467 | 15.2 | 0.31 |
| | CUT3R (Wang et al., 2025b) | Stream | 1.029 | 23.8 | 0.103 | 88.5 | 0.122 | 85.5 | 16.58 |
| | STREAM3R$^\alpha$ | Stream | 1.041 | 21.0 | 0.084 | 94.4 | 0.234 | 57.6 | 23.48 |

Table 7: 3D reconstruction comparison on NRGBD (Azinović et al., 2022). Our proposed method consistently achieves superior performance compared to optimization-based (Optim), streaming-based (Stream), and even full attention (FA) methods. STREAM3R$^\beta$-FA indicates adopting full attention in our trained model for 3D reconstruction.

| Method | Type | Acc↓ Mean | Acc↓ Med. | Comp↓ Mean | Comp↓ Med. | NC↑ Mean | NC↑ Med. |
|---|---|---|---|---|---|---|---|
| VGG-T (Wang et al., 2025a) | FA | 0.073 | 0.018 | 0.077 | 0.021 | 0.910 | 0.990 |
| DUSt3R-GA (Wang et al., 2024d) | Optim | 0.144 | 0.019 | 0.154 | 0.018 | 0.870 | 0.982 |
| MASt3R-GA (Leroy et al., 2024) | Optim | 0.085 | 0.033 | 0.063 | 0.028 | 0.794 | 0.928 |
| MonST3R-GA (Zhang et al., 2024a) | Optim | 0.272 | 0.114 | 0.287 | 0.110 | 0.758 | 0.843 |
| STREAM3R$^\beta$-FA | Stream | 0.057 | 0.014 | 0.028 | 0.013 | 0.910 | 0.993 |
| Spann3R (Wang & Agapito, 2024) | Stream | 0.416 | 0.323 | 0.417 | 0.285 | 0.684 | 0.789 |
| CUT3R (Wang et al., 2025b) | Stream | 0.099 | 0.031 | 0.076 | 0.026 | 0.837 | 0.971 |
| StreamVGGT (Zhuo et al., 2025) | Stream | 0.084 | 0.044 | 0.074 | 0.041 | 0.861 | 0.986 |
| VGG-T [Streaming] (Wang et al., 2025a) | Stream | 0.219 | 0.102 | 0.212 | 0.105 | 0.797 | 0.936 |
| STREAM3R$^\beta$ | Stream | 0.065 | 0.017 | 0.034 | 0.014 | 0.900 | 0.991 |

**3D Reconstruction on NRGBD.** We further include the comparison on NRGBD benchmark (Azinović et al., 2022) in Tab. 7. Here, we also include the comparison with a concurrent work StreamVGGT (Zhuo et al., 2025), which fine-tunes VGG-T into streaming version similar to our method. We also include VGG-T[streaming], which indicates using VGG-T in the streaming setting by replace the full attention in VGG-T into the causal attention. As can be seen, our method clearly outperforms all optimization-based and online methods, including the official VGG-T model. Direct use of VGG-T in the streaming setting substantially degrades performance, underscoring the need for fine-tuning under causal constraints. We also include STREAM3R$^\beta$-FA for comparison, which indicates replacing the causal attention in STREAM3R$^\beta$ into full attention (FA). Interestingly, STREAM3R$^\beta$-FA yields comparable performance compared to VGG-T and even better results on the completion metric. This highlights the effectiveness and generality of our proposed method.

**Camera Pose Estimation.** Following CUT3R (Wang et al., 2025b), we evaluate camera pose estimation accuracy on the Sintel (Butler et al., 2012), TUM-dynamics (Sturm et al.), and ScanNet (Dai et al., 2017) datasets. Sintel and TUM-dynamics both feature substantial dynamic motion, posing significant challenges to conventional SfM and SLAM pipelines. We report Absolute Translation

Table 8: Camera pose evaluation on Sintel (Butler et al., 2012), TUM-dynamic (Sturm et al.), and ScanNet (Dai et al., 2017) datasets. Our method achieves comparable performance with CUT3R on most benchmarks.

| Method | Type | Sintel | | | TUM-dynamics | | | ScanNet | | |
| --- | --- | --- | --- | --- | --- | --- | --- | --- | --- | --- |
| | | ATE ↓ | RPE trans ↓ | RPE rot ↓ | ATE ↓ | RPE trans ↓ | RPE rot ↓ | ATE ↓ | RPE trans ↓ | RPE rot ↓ |
| Particle-SfM (Zhao et al., 2022) | Optim | 0.129 | 0.031 | **0.535** | - | - | - | 0.136 | 0.023 | 0.836 |
| Robust-CVD (Kopf et al., 2021) | Optim | 0.360 | 0.154 | 3.443 | 0.153 | 0.026 | 3.528 | 0.227 | 0.064 | 7.374 |
| CasualSAM (Zhang et al., 2022) | Optim | 0.141 | **0.035** | 0.615 | 0.071 | **0.010** | 1.712 | 0.158 | 0.034 | 1.618 |
| DUSt3R-GA (Wang et al., 2024d) | Optim | 0.417 | 0.250 | 5.796 | 0.083 | 0.017 | 3.567 | 0.081 | 0.028 | 0.784 |
| MASt3R-GA (Leroy et al., 2024) | Optim | 0.185 | 0.060 | 1.496 | **0.038** | 0.012 | **0.448** | 0.078 | 0.020 | **0.475** |
| MonST3R-GA (Zhang et al., 2024a) | Optim | **0.111** | 0.044 | 0.869 | 0.098 | 0.019 | 0.935 | 0.077 | **0.018** | 0.529 |
| DUSt3R (Wang et al., 2024d) | Stream | 0.290 | 0.132 | 7.869 | 0.140 | 0.106 | 3.286 | 0.246 | 0.108 | 8.210 |
| Spann3R (Wang & Agapito, 2024) | Stream | 0.329 | 0.110 | 4.471 | 0.056 | 0.021 | 0.591 | 0.096 | 0.023 | 0.661 |
| CUT3R (Wang et al., 2025b) | Stream | **0.213** | **0.066** | **0.621** | 0.046 | 0.015 | 0.473 | 0.099 | 0.022 | **0.600** |
| STREAM3R$^\beta$ | Stream | **0.213** | 0.076 | 0.868 | **0.026** | **0.013** | **0.330** | **0.052** | **0.021** | 0.850 |

Error (ATE), Relative Translation Error (RPE$_{trans}$), and Relative Rotation Error (RPE$_{rot}$) after Sim(3) alignment with the ground truth, following the protocol in (Teed & Deng, 2021; Zhang et al., 2024a; Wang et al., 2025b). Our approach operates without requiring camera calibration, similar to the compared baselines (Teed & Deng, 2021). While many prior methods (Kopf et al., 2021; Zhang et al., 2022) address this via test-time optimization, which jointly estimates intrinsics and dense depth for each sequence. We focus on purely online processing. Tab. 8 reports results for Streaming (Stream) and Optimization (Optim) categories, with DUSt3R (Wang et al., 2024d) included in the latter (aligning all frames to the first frame without global alignment). Although optimization-based systems still achieve the lowest errors overall, our method establishes the strongest performance among streaming approaches, and notably surpasses CUT3R (Wang et al., 2025b) on both TUM-dynamics and ScanNet, demonstrating particular robustness in dynamic environments.

**3D Reconstruction on ETH3D.** To further verify performance on large-scale data with longer sequences, we include 3D reconstruction experiments on the ETH3D (Schöps et al., 2017) dataset, as shown in Tab. 9. As can be seen, global alignment (GA)-based methods (DUSt3R, MASt3R) perform significantly worse than feed-forward reconstruction methods (CUT3R and Ours), indicating that they struggle to generalize to challenging scenes and long video sequences. Furthermore, our method significantly outperforms other streaming approaches (CUT3R, Spann3R, SLAM3R). While the full-attention offline method VGGT performs strongly, our streaming method achieves the best Completeness score among all methods (0.245 vs. VGGT 0.305) and remains competitive in accuracy.

Table 9: 3D Reconstruction Comparison on ETH3D (Schöps et al., 2017). Our proposed method achieves competitive performance compared to optimization-based (Optim), streaming-based (Stream), and full attention (FA) methods.

| Method | Type | Acc↓ | | Comp↓ | | NC↑ | |
| --- | --- | --- | --- | --- | --- | --- | --- |
| | | Mean | Med. | Mean | Med. | Mean | Med. |
| DUSt3R-GA (Wang et al., 2024d) | Optim | 2.582 | 2.034 | 2.126 | 1.544 | 0.548 | 0.573 |
| MASt3R-GA (Leroy et al., 2024) | Optim | 2.682 | 2.458 | 2.206 | 1.734 | 0.531 | 0.540 |
| Fast3R (Yang et al., 2025) | FA | 0.832 | 0.691 | 0.978 | 0.683 | 0.667 | 0.766 |
| VGG-T (Wang et al., 2025a) | FA | **0.280** | **0.185** | 0.305 | 0.182 | **0.853** | **0.950** |
| CUT3R (Wang et al., 2025b) | Stream | 0.617 | 0.525 | 0.747 | 0.579 | 0.754 | 0.848 |
| Spann3R (Wang & Agapito, 2024) | Stream | 1.730 | 1.107 | 1.373 | 0.742 | 0.545 | 0.634 |
| SLAM3R (Liu et al., 2024) | Stream | 1.678 | 1.288 | 0.996 | 0.499 | 0.615 | 0.681 |
| STREAM3R$^\beta$ | Stream | **0.363** | **0.227** | **0.245** | **0.094** | **0.812** | **0.943** |

**Comparison with VGGT-SLAM.** We compare our method with VGGT-SLAM (Maggio et al., 2025) on both static scenes (NRGBD) and dynamic scenes (Sintel and TUM-dynamics). As shown in Tab. 11, our approach performs on par with SLAM-specialized techniques for static scene reconstruction. Moreover, Tab. 12 and Fig. 6 demonstrates that our method can robustly reconstruct dynamic scenes, a capability that conventional SLAM-based methods typically lack.

We further emphasize that our method targets a different problem setting from SLAM-based approaches. Our goal is to develop a unified streaming 3D/4D reconstruction pipeline capable of handling both (dynamic) foreground and background regions, whereas SLAM-based methods primarily focus on reconstructing static backgrounds and estimating accurate camera poses.

Despite these differing objectives, our approach is fully compatible with feed-forward SLAM systems and can be seamlessly integrated into their pipelines. As demonstrated in a recent work SLAM-Former (Yuan et al., 2025), streaming-based 3D reconstruction with KV caching can effectively support frontend tasks such as keyframe selection, tracking, and mapping within a SLAM system.

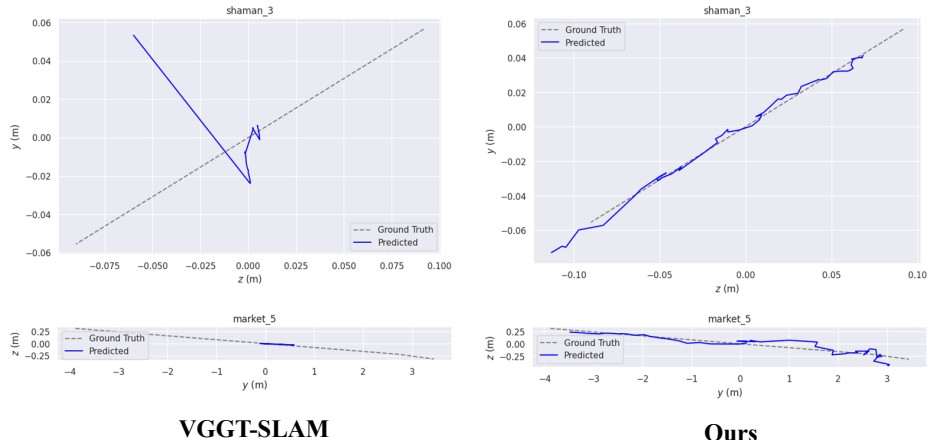

**VGGT-SLAM**  **Ours**

Figure 6: Visualizations of camera pose prediction on the dynamic Sintel dataset compared with VGGT-SLAM. As shown, our method demonstrates robustness in dynamic view reconstruction where static view consistency is not maintained, highlighting a capability that conventional SLAM-based methods typically lack.

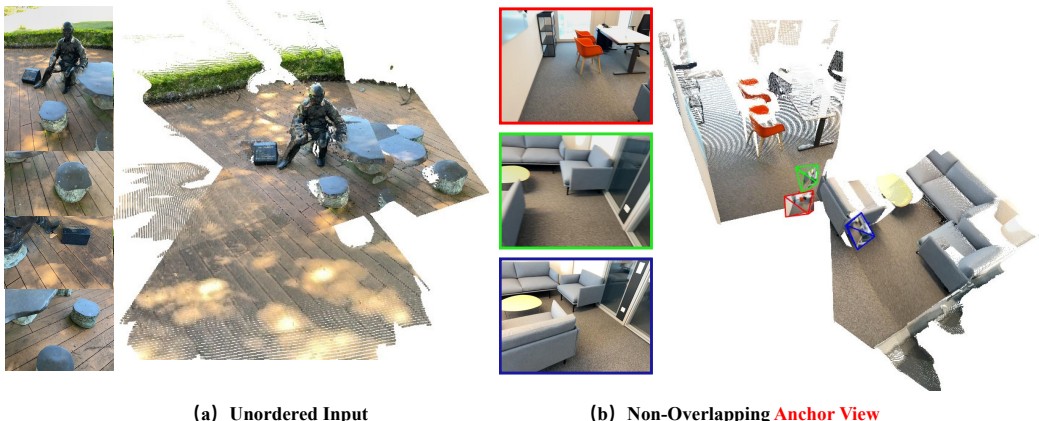

(a) **Unordered Input**   (b) **Non-Overlapping Anchor View**

Figure 7: Visualizations of input permutation and non-overlapping anchoring views. (a) STREAM3R maintains high accuracy under unordered input sequences. (b) STREAM3R successfully reconstructs the scene even when the anchoring view has no overlap with the rest of the sequence.

Table 10: Impact of first-view degradation on 3D reconstruction (7-Scenes). We compare the robustness of different methods against input degradation. The values in parentheses indicate the performance drop compared to the clean setting, marked in red.

| Method | Acc (Mean) ↓ | Acc (Med.) ↓ | Comp (Mean) ↓ | Comp (Med.) ↓ | NC (Mean) ↑ | NC (Med.) ↑ |
|---|---|---|---|---|---|---|
| VGGT (Wang et al., 2025a) | 0.087 | 0.039 | 0.091 | 0.039 | 0.787 | 0.890 |
| CUT3R (Wang et al., 2025b) | 0.126 | 0.047 | 0.154 | 0.031 | 0.727 | 0.834 |
| STREAM3R$^\beta$ | 0.122 | 0.044 | 0.101 | 0.038 | 0.746 | 0.856 |
| VGGT (w/ 1st view deg.) | 0.144 (+0.057) | 0.062 (+0.023) | 0.172 (+0.081) | 0.060 (+0.021) | 0.708 (-0.079) | 0.811 (-0.079) |
| CUT3R (w/ 1st view deg.) | 0.335 (+0.209) | 0.270 (+0.223) | 0.320 (+0.166) | 0.276 (+0.245) | 0.666 (-0.061) | 0.752 (-0.082) |
| STREAM3R$^\beta$ (w/ 1st view deg.) | 0.223 (+0.101) | 0.117 (+0.073) | 0.214 (+0.113) | 0.139 (+0.101) | 0.695 (-0.051) | 0.789 (-0.067) |

Table 11: 3D reconstruction comparison on dense NRGBD ($\sim$150 frames). Our method achieves comparable performance to SLAM-based methods on static scenes.

| Method | Type | Acc↓ | | Comp↓ | | NC↑ | |
|---|---|---|---|---|---|---|---|
| | | Mean | Med. | Mean | Med. | Mean | Med. |
| VGGT-SLAM Maggio et al. (2025) | SLAM-based | **0.039** | **0.017** | 0.028 | 0.009 | **0.781** | **0.939** |
| STREAM3R$^\beta$ | Stream | 0.046 | 0.020 | **0.012** | **0.005** | 0.756 | 0.923 |

Table 12: Camera pose comparison with VGGT-SLAM on Sintel (Butler et al., 2012) and TUM-dynamics (Sturm et al.). Compared to VGGT-SLAM, which focuses on static scene reconstruction, STREAM3R$^\beta$ shows robust camera estimation performance on dynamic scenarios.

| Method | Type | Sintel | | | TUM-dynamics | | |
|---|---|---|---|---|---|---|---|
| | | ATE ↓ | RPE trans ↓ | RPE rot ↓ | ATE ↓ | RPE trans ↓ | RPE rot ↓ |
| VGGT-SLAM | SLAM-based | 0.305 | 0.082 | 4.140 | 0.041 | 0.014 | 0.879 |
| STREAM3R$^\beta$ | Stream | **0.213** | **0.076** | **0.868** | **0.026** | **0.013** | **0.330** |

**Comparison Local and Global Point Map Prediction.** Our method supports both world-point maps and local-point maps (from depth and intrinsics). By using the extrinsics predicted by the camera head, the local-point map can be further projected into the global coordinate frame. As shown in Tab. 13, the point cloud projected from the local stream achieves better performance than direct world-point prediction. This observation shows the advantage of decomposing the challenging task of global-point map estimation into simpler subproblems. This finding is consistent with insights reported in VGGT and MapAnything (Keetha et al., 2025).

Table 13: Comparison of direct world point prediction and local depth projection on ETH3D.

| Method | Acc (Mean) ↓ | Acc (Med.) ↓ | Comp (Mean) ↓ | Comp (Med.) ↓ | NC (Mean) ↑ | NC (Med.) ↑ |
|---|---|---|---|---|---|---|
| Ours (from Local) | **0.363** | 0.227 | **0.245** | **0.094** | **0.812** | **0.943** |
| Ours (from Global) | 0.449 | **0.215** | 0.280 | 0.131 | 0.809 | 0.929 |

**More Analysis on Window Attention.** Here we provide additional analysis of the window-attention configuration of STREAM3R-W (with window sizes 4/8/16/32/64) on the NRGBD dataset with long sequences. As shown in Table 14, we find a positive correlation between attention window size and 3D reconstruction quality. This exposes a controllable trade-off between reconstruction quality and memory usage, allowing users to adapt the model to their specific hardware constraints.

Table 14: 3D reconstruction comparison on dense NRGBD ($\sim$150 views) with different window size. The memory is reported as peak memory in GB.

| Method | Acc (Mean) ↓ | Acc (Med.) ↓ | Comp (Mean) ↓ | Comp (Med.) ↓ | NC (Mean) ↑ | NC (Med.) ↑ | Memory |
|---|---|---|---|---|---|---|---|
| STREAM3R$^\beta$-W[4] | 0.074 | 0.031 | 0.021 | 0.011 | 0.699 | 0.894 | 6.00 |
| STREAM3R$^\beta$-W[8] | 0.075 | 0.030 | 0.019 | 0.010 | 0.693 | 0.889 | 6.65 |
| STREAM3R$^\beta$-W[16] | 0.080 | 0.033 | 0.022 | 0.011 | 0.706 | 0.896 | 7.96 |
| STREAM3R$^\beta$-W[32] | 0.069 | 0.028 | 0.021 | 0.010 | 0.729 | 0.910 | 10.58 |
| STREAM3R$^\beta$-W[64] | 0.051 | 0.023 | 0.019 | 0.010 | 0.737 | 0.913 | 15.93 |
| STREAM3R$^\beta$-W[128] | 0.048 | 0.021 | 0.016 | 0.009 | 0.752 | 0.921 | 24.51 |
| STREAM3R$^\beta$ (Causal) | **0.046** | **0.020** | **0.012** | **0.005** | **0.756** | **0.923** | 30.30 |

**Reconstruction Results on Longer Sequences.** To further evaluate long-sequence performance, we conduct experiments on NRGBD and 7-Scenes datasets using frame intervals of 2, 5, 7, 10, 20, and 40. As demonstrated in Tab. 15, our method consistently outperforms the baselines across all frame intervals and datasets, showing robust scalability to varying sequence lengths. We also report the performance of STREAM3R-W on a long trajectory of approximately 1.5K frames in Tab. 16. As can be seen, our method achieves substantially better performance than CUT3R on this challenging long-sequence setting, further demonstrating the advantages of our streaming design.

Table 15: 3D reconstruction comparison on longer sequences (NRGBD & 7-Scenes) with different intervals. We report the median metrics. The interval indicates the sampling sparsity, and the approximate number of input views is shown in parentheses.

| | NRGBD Dataset | | | | | | 7-Scenes Dataset | | | | | |
|---|---|---|---|---|---|---|---|---|---|---|---|---|
| Interval (Views) | 40 ($\sim$35) | 20 ($\sim$75) | 10 ($\sim$150) | 7 ($\sim$210) | 5 ($\sim$370) | 2 ($\sim$750) | 40 ($\sim$30) | 20 ($\sim$60) | 10 ($\sim$125) | 7 ($\sim$140) | 5 ($\sim$250) | 2 ($\sim$500) |
| *Accuracy ↓* | | | | | | | | | | | | |
| CUT3R (Wang et al., 2025b) | 0.032 | 0.042 | 0.064 | 0.110 | 0.179 | 0.266 | 0.013 | 0.013 | 0.019 | 0.039 | 0.087 | 0.161 |
| Spann3R (Wang & Agapito, 2024) | 0.074 | 0.068 | 0.100 | 0.118 | 0.136 | 0.104 | 0.139 | 0.074 | 0.051 | 0.056 | 0.084 | 0.104 |
| SLAM3R (Liu et al., 2024) | 0.113 | 0.107 | 0.109 | 0.117 | 0.119 | 0.113 | 0.106 | 0.096 | 0.100 | 0.094 | 0.097 | 0.111 |
| STREAM3R$^\beta$ | **0.019** | **0.019** | **0.020** | **0.022** | **0.025** | **0.028** | **0.013** | **0.012** | **0.010** | **0.015** | **0.021** | **0.021** |
| *Completeness ↓* | | | | | | | | | | | | |
| CUT3R (Wang et al., 2025b) | 0.013 | 0.010 | 0.011 | 0.034 | 0.083 | 0.134 | 0.011 | 0.008 | 0.008 | 0.013 | 0.048 | 0.066 |
| Spann3R (Wang & Agapito, 2024) | 0.033 | 0.023 | 0.031 | 0.045 | 0.041 | 0.065 | 0.089 | 0.048 | 0.015 | 0.017 | 0.041 | 0.065 |
| SLAM3R (Liu et al., 2024) | 0.046 | 0.027 | 0.015 | 0.021 | 0.012 | 0.072 | 0.053 | 0.031 | 0.019 | 0.015 | 0.032 | 0.056 |
| STREAM3R$^\beta$ | **0.008** | **0.006** | **0.005** | **0.009** | **0.016** | **0.018** | **0.012** | **0.009** | **0.006** | **0.009** | **0.015** | **0.021** |
| *Normal Consistency (NC) ↑* | | | | | | | | | | | | |
| CUT3R (Wang et al., 2025b) | 0.943 | 0.908 | 0.825 | 0.726 | 0.686 | 0.638 | 0.806 | 0.750 | 0.693 | 0.602 | 0.595 | 0.573 |
| Spann3R (Wang & Agapito, 2024) | 0.750 | 0.724 | 0.657 | 0.624 | 0.611 | 0.570 | 0.710 | 0.699 | 0.641 | 0.571 | 0.518 | 0.538 |
| SLAM3R (Liu et al., 2024) | 0.764 | 0.729 | 0.686 | 0.693 | 0.637 | 0.625 | 0.655 | 0.623 | 0.590 | 0.569 | 0.609 | 0.576 |
| STREAM3R$^\beta$ | **0.976** | **0.958** | **0.923** | **0.867** | **0.792** | **0.765** | **0.830** | **0.773** | **0.712** | **0.662** | **0.648** | **0.622** |

Table 16: 3D reconstruction comparison on thousands of frames ($\sim$1.5k). Our method demonstrates superior stability on extremely long sequences compared to CUT3R.

| Method | Acc (Mean) ↓ | Acc (Med.) ↓ | Comp (Mean) ↓ | Comp (Med.) ↓ | NC (Mean) ↑ | NC (Med.) ↑ |
|---|---|---|---|---|---|---|
| CUT3R (Wang et al., 2025b) | 0.411 | 0.315 | 0.224 | 0.146 | 0.544 | 0.581 |
| STREAM3R$^\beta$-W[16] | **0.094** | **0.039** | **0.028** | **0.015** | **0.627** | **0.716** |

## A.5 LIMITATIONS

Our method comes with some limitations. First, the naïve causal modeling naturally suffers from error accumulation and drifting (Zhang & Agrawala, 2025). Some inference strategies can be proposed to alleviate this issue. Second, currently STREAM3R is still a regression model with deterministic outputs. Extending it further into an autoregressive generative model (Chen et al., 2025; Zhang & Agrawala, 2025) shall further unlock a series of downstream applications. Finally, since STREAM3R follows a similar design of modern LLMs, more training techniques like MLA (DeepSeek-AI et al., 2024) can be introduced to further boost the training efficiency and performance.

