# OpenReview forum: "STream3R: Scalable Sequential 3D Reconstruction with Causal Transformer"
_ICLR.cc/2026/Conference — ICLR 2026 Poster_

### Official Review · Reviewer_shrJ · 2025-10-27

**Soundness:** 3
**Presentation:** 3
**Contribution:** 2
**Rating:** 4
**Confidence:** 5

**Summary:**

This paper introduces a streaming version of VGGT, which can process image sequences efficiently using causal attention.
The authors conduct extensive experiments to show the effectiveness and advantages of their method on various tasks.

**Strengths:**

1. This paper is well-written and easy to follow. It also provides a nice demo to show their performance.
2. The authors conduct extensive experiments to show the effectiveness of their method.

**Weaknesses:**

1. The backbone is highly related to that of VGGT, and the usage of causal attention do not have enough contribution considering the high computational cost when the input sequences are very long (same with VGGT).
2. I think that since the authors highlighted the advantage of Stream3R-W in Table 4 when analyzing memory usage, its results should also be included in other performance experiments (like reconstruction) for comparison. After all, the memory usage of Stream3R without the window mechanism gradually exceeds that of CUT3R (also a streaming-based method). I believe that a comprehensive analysis of Stream3R and Stream3R-W would help provide a better evaluation of this paper.
3. Reconstruction results on longer sequences. The sequences used in Table 3 are too short, which is not enough to evaluate a streaming-based method. Please provide more comparison with Spann3R, SLAM3R, and CUT3R on sequences at the intervals of 2, 10, 20, 40 (7-scenes and NRGBD).
4. Please provide reconstruction results comparison on ETH3D datasets, like VGGT.

Please answer the above questions carefully and provide more thorough discussion and comparison. I will consider raising the score.

**Questions:**

Please refer to the Weaknesses.

---

> ### Author Response · Authors · 2025-11-25
> **Rebuttal by Authors (Part I)**
>
> We thank the reviewer for finding our paper well-written and easy to follow, and for appreciating our extensive experiments and demo. We have also carefully addressed the concerns and provide more analytics below.
>
> ---
>
> ### W1：Efficiency and Contribution of Causal Attention
>
> Thanks for the question. Our framework is specifically designed for streaming 3D reconstruction, a setting where computational and memory efficiency are critical and largely underexplored by existing methods. Prior approaches such as VGGT rely on full-attention architectures tailored to short multi-view inputs while their memory and compute grow with the number of frames. Instead of introducing unnecessary architectural complexity, we aim to develop a clean and unified design consistent with the decoder-only transformer paradigm widely adopted in modern LLMs.
>
> This design enables true streaming inference, where frames are processed sequentially under a fixed compute and memory budget while still maintaining a consistent 3D representation over long sequences. In particular, our variant STream3R-W, which integrates windowed attention into this streaming 3D reconstruction pipeline, achieves high-quality 3D reconstruction over long sequences with fixed resource usage (Tab. 4), while also attaining the highest FPS among compared methods (Tab. 2)  without sacrificing, while even improving performance on video depth estimation (Bonn dataset). This demonstrates that our LLM-inspired, windowed decoder design offers a practically important advantage over VGGT-style full-attention models and points to a novel and promising direction for efficient streaming 3D perception.
>
> Furthermore, compared to the existing streaming method CUT3R, the ablation study in Sec. 5.3 demonstrates that our approach converges significantly faster with 60% speedup in training. This further validates the effectiveness of our decoder-only transformer over existing RNN-based streaming designs.
>
> ---
>
> ### W2: More Analysis on Window Attention
>
> Following the reviewers' suggestions, here we provide additional analysis of the window-attention configuration of STream3R-W (with window sizes 4/8/16/32/64) on the NRGBD dataset with long sequences. As shown in the Tab. A below, we find a positive correlation between attention window size and 3D reconstruction quality. This exposes a controllable trade-off between reconstruction quality and memory usage, allowing users to adapt the model to their specific hardware constraints.
>
> In addition, we report the performance of STream3R-W on a extremely long trajectory of approximately 1.5K frames in Tab. B. As can be seen, our method achieves substantially better performance than CUT3R on this challenging long-sequence setting, further demonstrating the advantages of our streaming design.
>
> ***Table A: 3D Reconstruction Comparison on Dense NRGBD (~150 views) with Different Window Size.***
>
> | Method | Acc (Mean) ↓ | Acc (Med.) ↓ | Comp (Mean) ↓ | Comp (Med.) ↓ | NC (Mean) ↑ | NC (Med.) ↑ | Peak Memory (GB) |
> | :--- | :---: | :---: | :---: | :---: | :---: | :---: | :---: |
> | **Ours-W[4]** | 0.074 | 0.031 | 0.021 | 0.011 | 0.699 | 0.894 | 6.00 |
> | **Ours-W[8]** | 0.075 | 0.030 | 0.019 | 0.010 | 0.693 | 0.889 | 6.65 |
> | **Ours-W[16]** | 0.080 | 0.033 | 0.022 | 0.011 | 0.706 | 0.896 | 7.96 |
> | **Ours-W[32]** | 0.069 | 0.028 | 0.021 | 0.010 | 0.729 | 0.910 | 10.58 |
> | **Ours-W[64]** | 0.051 | 0.023 | 0.019 | 0.010 | 0.737 | 0.913 | 15.93 |
> | **Ours-W[128]** | 0.048 | 0.021 | 0.016 | 0.009 | 0.752 | 0.921 | 24.51 |
> | **Ours (Full Causal Context)** | **0.046** | **0.020** | **0.012** | **0.005** | **0.756** | **0.923** | 30.30 |
>
> ***Table B: 3D Reconstruction Comparison on Thousands of Frames (~1.5k)***
>
> | Method | Acc (Mean) ↓ | Acc (Med.) ↓ | Comp (Mean) ↓ | Comp (Med.) ↓ | NC (Mean) ↑ | NC (Med.) ↑ |
> | :--- | :---: | :---: | :---: | :---: | :---: | :---: |
> | **CUT3R** | 0.411 | 0.315 | 0.224 | 0.146 | 0.544 | 0.581 |
> | **Ours-W[16]** | **0.094** | **0.039** | **0.028** | **0.015** | **0.627** | **0.716** |

---

> > ### Author Response · Authors · 2025-11-25
> > **Rebuttal by Authors (Part II)**
> >
> > ### W3: Reconstruction results on longer sequences
> >
> > Thanks for the suggestions. To further evaluate long-sequence performance, we include experiments on NRGBD and 7-Scenes dataset using intervals of *2, 5, 7, 10, 20, and 40* (given that the number of input views is inversely proportional to the frame interval, we evaluated additional intervals beyond requested to provide a finer-grained analysis of the trend). As can be seen, across all frame intervals and datasets, our method consistently outperforms the baselines.
> >
> > ***Table C: 3D Reconstruction Comparison on NRGBD with Different Intervals (Median Metric).***
> >
> > *Accuracy ↓*
> > | Interval (Num of Views): | 40 (~35) | 20 (~75) | 10 (~150) | 7 (~210) | 5 (~370) | 2 (~750) |
> > | :--- | :---: | :---: | :---: | :---: | :---: | :---: |
> > | CUT3R | 0.032 | 0.042 | 0.064 | 0.110 | 0.179 | 0.266 |
> > | Spann3R | 0.074 | 0.068 | 0.100 | 0.118 | 0.136 | 0.104 |
> > | SLAM3R | 0.113 | 0.107 | 0.109 | 0.117 | 0.119 | 0.113 |
> > | **Ours** | **0.019** | **0.019** | **0.020** | **0.022** | **0.025** | **0.028** |
> >
> > *Completeness ↓*
> > | Interval (Num of Views): | 40 (~35) | 20 (~75) | 10 (~150) | 7 (~210) | 5 (~370) | 2 (~750) |
> > | :--- | :---: | :---: | :---: | :---: | :---: | :---: |
> > | CUT3R | 0.013 | 0.010 | 0.011 | 0.034 | 0.083 | 0.134 |
> > | Spann3R | 0.033 | 0.023 | 0.031 | 0.045 | 0.041 | 0.065 |
> > | SLAM3R | 0.046 | 0.027 | 0.015 | 0.021 | 0.012 | 0.072 |
> > | **Ours** | **0.008** | **0.006** | **0.005** | **0.009** | **0.016** | **0.018** |
> >
> > *NC ↑*
> > | Interval (Num of Views): | 40 (~35) | 20 (~75) | 10 (~150) | 7 (~210) | 5 (~370) | 2 (~750) |
> > | :--- | :---: | :---: | :---: | :---: | :---: | :---: |
> > | CUT3R | 0.943 | 0.908 | 0.825 | 0.726 | 0.686 | 0.638 |
> > | Spann3R | 0.750 | 0.724 | 0.657 | 0.624 | 0.611 | 0.570 |
> > | SLAM3R | 0.764 | 0.729 | 0.686 | 0.693 | 0.637 | 0.625 |
> > | **Ours** | **0.976** | **0.958** | **0.923** | **0.867** | **0.792** | **0.765** |
> >
> >
> > ***Table D: 3D Reconstruction Comparison on 7-Scenes with Different Intervals (Median Metric).***
> >
> > *Accuracy ↓*
> > | Interval (Num of Views): | 40 (~30) | 20 (~60) | 10 (~125) | 7  (~140) | 5  (~250) | 2 (~500) |
> > | :--- | :---: | :---: | :---: | :---: | :---: | :---: |
> > | CUT3R | 0.013 | 0.013 | 0.019 | 0.039 | 0.087 | 0.161 |
> > | Spann3R | 0.139 | 0.074 | 0.051 | 0.056 | 0.084 | 0.104 |
> > | SLAM3R | 0.106 | 0.096 | 0.100 | 0.094 | 0.097 | 0.111 |
> > | **Ours** | **0.013** | **0.012** | **0.010** | **0.015** | **0.021** | **0.021** |
> >
> > *Completeness ↓*
> > | Interval (Num of Views): | 40 (~30) | 20 (~60) | 10 (~125) | 7  (~140) | 5  (~250) | 2 (~500) |
> > | :--- | :---: | :---: | :---: | :---: | :---: | :---: |
> > | CUT3R | 0.011 | 0.008 | 0.008 | 0.013 | 0.048 | 0.066 |
> > | Spann3R | 0.089 | 0.048 | 0.015 | 0.017 | 0.041 | 0.065 |
> > | SLAM3R | 0.053 | 0.031 | 0.019 | 0.015 | 0.032 | 0.056 |
> > | **Ours** | **0.012** | **0.009** | **0.006** | **0.009** | **0.015** | **0.021** |
> >
> > *NC ↑*
> > | Interval (Num of Views): | 40 (~30) | 20 (~60) | 10 (~125) | 7  (~140) | 5  (~250) | 2 (~500) |
> > | :--- | :---: | :---: | :---: | :---: | :---: | :---: |
> > | CUT3R | 0.806 | 0.750 | 0.693 | 0.602 | 0.595 | 0.573 |
> > | Spann3R | 0.710 | 0.699 | 0.641 | 0.571 | 0.518 | 0.538 |
> > | SLAM3R | 0.655 | 0.623 | 0.590 | 0.569 | 0.609 | 0.576 |
> > | **Ours** | **0.830** | **0.773** | **0.712** | **0.662** | **0.648** | **0.622** |
> >
> >
> > ---
> >
> > ### W4: 3D Reconstruction on ETH3D
> >
> > We also provide 3D reconstruction results on the ETH3D dataset in the table below. As can be seen, our method outperforms the streaming baselines by a clear margin. While the full-attention offline method VGGT performs great, our streaming method achieves the best Comp score among all methods (0.245 vs VGGT 0.305) and remains competitive in accuracy.
> >
> > ***Table E: 3D Reconstruction Comparison on ETH3D.***
> >
> > | Method | Type | Acc ↓ (Mean / Med.) | Comp ↓ (Mean / Med.) | NC ↑ (Mean / Med.) |
> > | :--- | :---: | :---: | :---: | :---: |
> > | DUSt3R-GA | Optim. | 2.582 / 2.034 | 2.126 / 1.544 | 0.548 / 0.573 |
> > | MASt3R-GA | Optim. | 2.682 / 2.458 | 2.206 / 1.734 | 0.531 / 0.540 |
> > | Fast3R | FA | 0.832 / 0.691 | 0.978 / 0.683 | 0.667 / 0.766 |
> > | VGGT | FA | **0.280** / **0.185** | 0.305 / 0.182 | **0.853** / **0.950** |
> > |
> > | CUT3R | Stream. | 0.617 / 0.525 | 0.747 / 0.579 | 0.754 / 0.848 |
> > | Span3R | Stream. | 1.730 / 1.107 | 1.373 / 0.742 | 0.545 / 0.634 |
> > | SLAM3R | Stream. | 1.678 / 1.288 | 0.996 / 0.499 | 0.615 / 0.681 |
> > | **Ours** | Stream. | **0.363** / **0.227** | **0.245** / **0.094** | **0.812** / **0.943** |
> >
> > ---
> >
> > We hope our responses have addressed the reviewer's concerns. If anything remains unclear or if you have further questions, we'd be happy to clarify and discuss them in more detail.

---

### Official Review · Reviewer_vTS9 · 2025-10-31

**Soundness:** 3
**Presentation:** 3
**Contribution:** 3
**Rating:** 8
**Confidence:** 4

**Summary:**

Conducting feed-forward 3D reconstruction from monocular videos is nowadays a very hot topic and a series of works followed dust3R and VGGT. This work is an extension of VGGT to make reconstruction be able to support streaming input. The goal is reached by reformulating the traditional global attention of traditional vggt framework into a sequential attention mechanism. Experiments are sufficient to verify the superior performance of the proposed framework.

**Strengths:**

The target problem to support feed-forward and streaming 3D reconstruction is timely, the proposed framework also makes sense. Assume the review does not need to consider concurrent works, I think the proposed framework is compatable with existing concurrent works in terms of both performance and novelty.

**Weaknesses:**

I only have some minor concerns:
- As shown in table 7, StreamVGGT is attending the comparison which is a concurrent work with similiar key idea. From the results, it seems the proposed method outperforms StreamVGGT, but with no any explanations. I understand this work is a concurrent work. However, the explanation about the differences bettween streamVGGT and the proposed method and why the results of the proposed method are better is helpful for understanding.

- In the contribution list, there are 2 points are highlighted: 1) compatible with LLM-style training, allowding efficient and scalable context accumulation across frames; 2) supports both world- and local- pointmap and natually generallizes to large-scale novel view synthesis. However, there is no any experiments and discussions to support this.

**Questions:**

No more.

---

> ### Author Response · Authors · 2025-11-25
> **Rebuttal by Authors**
>
> We are encouraged by the reviewer's recognition of the timeliness, soundness, and strong performance of our proposed framework. We also address specific questions in detail below.
>
> ---
>
> ### Q1: Discussion with StreamVGGT
>
> StreamVGGT is a **concurrent** work that also targets streaming 3D reconstruction using a causal-transformer architecture similar to ours. Following the reviewers' suggestions, we highlight the key differences below:
>
> 1. **Training Strategy**
>    We train end-to-end directly with ground-truth 3D supervision, whereas StreamVGGT relies on distillation from VGG-T outputs to stabilize training. In our experiments, we do not find distillation necessary and believe that this contributes to the stronger performance of STream3R compared to StreamVGGT, as shown in Tab. 7 of the appendix.
>
> 2. **Memory Growth in Streaming**
>    StreamVGGT adopts a causal transformer but does not address VRAM growth as more frames arrive during streaming. In contrast, we propose STream3R-W, which includes detailed analysis of its memory usage, speed, and reconstruction accuracy under fixed VRAM and fixed computational cost, achieving high-FPS, high-quality streaming 3D reconstruction.
>
> 3. **Metric-Scale Reconstruction**
>    We additionally introduce STream3R-α, a metric-scale reconstruction variant. StreamVGGT provides only scale-invariant depth predictions and does not support metric reconstruction.
>
> ---
>
> ### Q2: LLM-style Training, World/Local-pointmap, and Novel View Synthesis
>
> Thank you for the suggestions. We plan to include a more complete discussion and additional experiments on:
>
> - **LLM-Style Training Techniques**
>   Beyond inference-time KV-cache usage, we apply several LLM-inspired optimization techniques, such as:
>   - Triton-optimized RMSNorm and attention kernels [A],
>   - Bias-free FFN layers and an MLP ratio of 4, following GPT/LLaMA conventions,
>   - LLM-style learning rate and weight decay strategies [B].
>
> - **Local and Global Point-Map Prediction**
>    Our method supports both world-point maps and local-point maps (from depth and intrinsics). By using the extrinsics predicted by the camera head, the local-point map can be further projected into the global coordinate frame. As shown in the Tab. A below, the point cloud projected from the local stream achieves better performance than direct world-point prediction. This observation shows the advantage of decomposing the challenging task of global-point map estimation into simpler subproblems. This finding is consistent with insights reported in VGGT and MapAnything [C].
>
>    ***Table A: Comparison of Direct World-Point Prediction vs. Local-Point Projection on ETH3D***
>    | Method | Acc (Mean) ↓ | Acc (Med.) ↓ | Comp (Mean) ↓ | Comp (Med.) ↓ | NC (Mean) ↑ | NC (Med.) ↑ |
>    | :--- | :---: | :---: | :---: | :---: | :---: | :---: |
>    | *Ours (from Local)* | **0.363** | 0.227 | **0.245** | **0.094** | **0.812** | **0.943** |
>    | *Ours (from Global)* | 0.449 | **0.215** | 0.280 | 0.131 | 0.809 | 0.929 |
>    | |
>
> - **Novel View Synthesis (NVS)**
>    Thanks for the question. We utilize the dense point maps and camera poses predicted by our model as the geometric initialization for downstream view synthesis. Specifically, we export our results to the COLMAP format and use them to initialize and train 3D Gaussian Splatting models. This initialization allows for effective training on complex video sequences without requiring external structure-from-motion tools. As shown in Fig. 7 of the appendix, our reconstructions lead to high-quality NVS renderings.
>
> ---
>
> We hope our responses have addressed the reviewer's concerns. If anything remains unclear or if you have further questions, we'd be happy to clarify and discuss them in more detail.
>
> ---
> [A]: *Pin-Lun Hsu, et al. "Liger-Kernel: Efficient Triton Kernels for LLM Training." Championing Open-source DEvelopment in ML Workshop at ICML, 2025.*
> [B]: *Maksym Andriushchenko, et al. "Why Do We Need Weight Decay in Modern Deep Learning?" Advances in Neural Information Processing Systems, 2024.*
> [C]: *Nikhil Keetha, et al. "MapAnything: Universal Feed-Forward Metric 3D Reconstruction." arXiv preprint arXiv:2509.13414, 2025.*

---

> > ### Comment · Reviewer_vTS9 · 2025-11-28
> > **thanks for responses**
> >
> > Many thanks for the anwers. My questions have been addressed. I will keep my rating.

---

> > > ### Author Response · Authors · 2025-11-28
> > > **Official Comment by Authors**
> > >
> > > Dear reviewer vTS9, we sincerely appreciate your encouraging feedback. If anything requires further explanation, we are  happy to assist.

---

### Official Review · Reviewer_JPV9 · 2025-11-01

**Soundness:** 3
**Presentation:** 3
**Contribution:** 3
**Rating:** 6
**Confidence:** 5

**Summary:**

This paper introduces STREAM3R, a novel framework that reformulates sequential 3D reconstruction as a decoder-only causal transformer problem, inspired by autoregrssive-based LLMs. The core contribution is an efficient streaming architecture that processes frames sequentially, using causal attention to integrate geometric context from a growing KVCache of previously observed features. This design enables high scalability for long sequences and efficient inference by leveraging modern LLM-style techniques like windowed attention. The method achieves state-of-the-art or highly competitive performance on numerous 3D reconstruction and video depth benchmarks, outperforming prior streaming and even full-attention-based approaches.

**Strengths:**

1. The core architectural proposal to use a decoder-only causal transformer with a KVCache is a novel and highly effective paradigm for streaming 3D reconstruction. It offers a more scalable and powerful alternative to prior methods based on RNNs-like structure (like CUT3R) or expensive global optimization (like VGG-T or DUSt3R-GA).

2. The method demonstrates sota competitive performance across a comprehensive suite of benchmarks, including video depth estimation, 3D reconstruction, and camera pose estimation. It consistently and significantly outperforms its most direct streaming competitor, CUT3R.

3. The paper provides a thorough analysis of computational trade-offs. The STREAM3R-V-W[5] (windowed) variant is particularly noteworthy, achieving the fastest inference speed (32.9 FPS) among all streaming methods while simultaneously delivering top-tier reconstruction accuracy, demonstrating the practical viability of the approach. The memory usage analysis in Table 4 clearly validates the scalability claims.

**Weaknesses:**

please refer to the weakness part.

**Questions:**

1. The primary technical weakness is the inherent risk of error accumulation and drift common to online, sequential methods. The paper does not propose or evaluate explicit mechanisms to combat this (e.g., loop closure, keyframe-based optimization) and it is unclear how the system would perform on extremely long sequences (e.g., thousands of frames) beyond what was tested.

2. The global coordinate system is anchored to the very first frame using a learnable [reg] token. The robustness of this simple mechanism is not ablated or discussed. It is unclear how the system would perform if the initial frame is of poor quality (e.g., blurry, featureless, or heavily occluded), which could compromise the entire reconstruction.

3. A key claim is the compatibility with "LLM-style training infrastructure," but the experiments primarily validate LLM-style inference techniques (KVCache, windowed attention). The paper does not include experiments applying advanced LLM-style training optimizations (e.g., specific curriculum strategies, advanced data parallelism, or training-focused attention mechanisms, which the authors themselves allude to in the limitations). This makes the claim about training benefits less substantiated than the clear inference benefits, and complementary experiments would strengthen this claim.

---

> ### Author Response · Authors · 2025-11-25
> **Rebuttal by Authors (Part I)**
>
> We thank the reviewer for recognizing the novelty, scalability, and strong empirical performance of our proposed framework. We also appreciate the reviewer's insightful feedback and address specific concerns in detail below.
>
> ---
>
> ### Q1 (a): Error Accumulation and Performance on Long Sequences
>
> Thanks for the question. To demonstrate the performance of our proposed method on long sequences, we evaluate our method on N-RGBD with different intervals and compare with other streaming methods. While some degree of error accumulation is inevitable for all the streaming methods, our results in Tab.A show that the proposed method consistently outperforms the baselines across all sequence lengths.
>
> ***Table A: 3D Reconstruction Comparison on NRGBD with Different Intervals (Median Metric).***
>
> *Accuracy ↓*
> | Interval (Num of Views): | 40 (~35) | 20 (~75) | 10 (~150) | 7 (~210) | 5 (~370) | 2 (~750) |
> | :--- | :---: | :---: | :---: | :---: | :---: | :---: |
> | CUT3R | 0.032 | 0.042 | 0.064 | 0.110 | 0.179 | 0.266 |
> | Spann3R | 0.074 | 0.068 | 0.100 | 0.118 | 0.136 | 0.104 |
> | SLAM3R | 0.113 | 0.107 | 0.109 | 0.117 | 0.119 | 0.113 |
> | **Ours** | **0.019** | **0.019** | **0.020** | **0.022** | **0.025** | **0.028** |
>
> *Completeness ↓*
> | Interval (Num of Views): | 40 (~35) | 20 (~75) | 10 (~150) | 7 (~210) | 5 (~370) | 2 (~750) |
> | :--- | :---: | :---: | :---: | :---: | :---: | :---: |
> | CUT3R | 0.013 | 0.010 | 0.011 | 0.034 | 0.083 | 0.134 |
> | Spann3R | 0.033 | 0.023 | 0.031 | 0.045 | 0.041 | 0.065 |
> | SLAM3R | 0.046 | 0.027 | 0.015 | 0.021 | 0.012 | 0.072 |
> | **Ours** | **0.008** | **0.006** | **0.005** | **0.009** | **0.016** | **0.018** |
>
> *NC ↑*
> | Interval (Num of Views): | 40 (~35) | 20 (~75) | 10 (~150) | 7 (~210) | 5 (~370) | 2 (~750) |
> | :--- | :---: | :---: | :---: | :---: | :---: | :---: |
> | CUT3R | 0.943 | 0.908 | 0.825 | 0.726 | 0.686 | 0.638 |
> | Spann3R | 0.750 | 0.724 | 0.657 | 0.624 | 0.611 | 0.570 |
> | SLAM3R | 0.764 | 0.729 | 0.686 | 0.693 | 0.637 | 0.625 |
> | **Ours** | **0.976** | **0.958** | **0.923** | **0.867** | **0.792** | **0.765** |
>
> Following the reviewer's suggestion, we further evaluate our method on extremely long sequences consisting of approximately 1.5K frames from the NRGBD dataset. As shown in the Tab. B below, our approach substantially outperforms the streaming-based baseline CUT3R. In particular, we observe that CUT3R struggles to maintain reliable registration (i.e., drifting) when processing sequences exceeding 1K frames. For this experiment, we use STream3R-W with a window size of 16 to ensure constant memory consumption.
>
> ***Table B: 3D Reconstruction Comparison on Thousands of Frames (~1.5k)***
>
> | Method | Acc (Mean) ↓ | Acc (Med.) ↓ | Comp (Mean) ↓ | Comp (Med.) ↓ | NC (Mean) ↑ | NC (Med.) ↑ |
> | :--- | :---: | :---: | :---: | :---: | :---: | :---: |
> | **CUT3R** | 0.411 | 0.315 | 0.224 | 0.146 | 0.544 | 0.581 |
> | **Ours-W[16]** | **0.094** | **0.039** | **0.028** | **0.015** | **0.627** | **0.716** |
>
> ---
>
> ### Q1 (b): Loop-closure and Key-frame Detection
>
> Loop-closure and key-frame detection are standard components in SLAM systems for mitigating long-sequence drift. STream3R, however, is introduced as the first causal-transformer–based streaming 3D reconstruction method, with a primary focus on providing an efficient and scalable solution for downstream applications.
>
> These two lines of contribution is orthogonal and can be beneficial to each other. While we plan to incorporate loop-closure and key-frame mechanisms in future work, the current model already offers strong geometric priors that can directly benefit downstream SLAM pipelines, as also demonstrated in SLAM-Former [A] (i.e., the online stage of SLAM-former follows exactly the same design as Stream3R.)
>
> Also, SLAM-based methods are limited to modeling the static scene, while our proposed method can also be applied to the reconstruction of dynamic foreground, as shown in the newly added Tab. 11 and Tab. 12 in the appendix.

---

> ### Author Response · Authors · 2025-11-25
> **Rebuttal by Authors (Part II)**
>
> ### Q2: First Frame Robustness
>
> We would like to clarify that the [reg] token is common and implemented in a similar way as Fast3R and VGGT, which is a standard strategy and not the main contribution of our proposed framework.
>
> Moreover, using the first frame as the global coordinate system is a standard convention across DUSt3R and its follow-up works, including MASt3R, MonST3R, CUT3R, VGGT, and ours. Following the reviewer's suggestions, we follow the degradation pipeline of Real-ESRGAN [B] to corrupt the first frame of each sequence, and then evaluate VGGT, CUT3R, and our method on the 7-Scenes dataset quantitatively. As shown in Tab. C, all methods experience some degradation. However, CUT3R's Acc error increases significantly from 0.126 to 0.335, whereas STream3R only increases from 0.122 to 0.223, demonstrating that our method is considerably more robust under such challenging conditions.
>
> ***Table C: Impact of First-View Degradation on 3D Reconstruction (7-Scenes).***
>
> | Method | Acc (Mean) ↓ | Acc (Med.) ↓ | Comp (Mean) ↓ | Comp (Med.) ↓ | NC (Mean) ↑ | NC (Med.) ↑ |
> | :--- | :---: | :---: | :---: | :---: | :---: | :---: |
> | **VGGT** | 0.087 | 0.039 | 0.091 | 0.039 | 0.787 | 0.890 |
> | **CUT3R** | 0.126 | 0.047 | 0.154 | 0.031 | 0.727 | 0.834 |
> | **Ours** | 0.122 | 0.044 | 0.101 | 0.038 | 0.746 | 0.856 |
> | **VGGT** (w/ 1st view deg.) | 0.144 (+0.057) | 0.062 (+0.023) | 0.172 (+0.081) | 0.060 (+0.021) | 0.708 (-0.079) | 0.811 (-0.079) |
> | **CUT3R** (w/ 1st view deg.) | 0.335 (+0.209) | 0.270 (+0.223) | 0.320 (+0.166) | 0.276 (+0.245) | 0.666 (-0.061) | 0.752 (-0.082) |
> | **Ours** (w/ 1st view deg.) | 0.223 (+0.101) | 0.117 (+0.073) | 0.214 (+0.113) | 0.139 (+0.101) | 0.695 (-0.051) | 0.789 (-0.067) |
>
> Furthermore, as shown in the newly added Fig. 5 (b) in the appendix, even when the first frame has very little overlap, our model still shows strong implicit relative pose-learning capability for the other views.
>
> ---
>
> ### Q3: LLM Training Claim
>
> Beyond using KV-cache during inference, we also adopt several LLM-oriented training techniques:
>
> - We use Triton-based RMSNorm and Attention kernels [C], optimized for large-model training.
> - Following standard LLM design (e.g., GPT, LLaMA), we:
>   - remove the bias terms in FFN layers,
>   - set the MLP ratio to 4,
>   - and apply similar learning rate and weight decay schedules [D] commonly used to stabilize and improve convergence.
>
> In future work, we plan to extend the current attention module to GQA [E] and MLA [F] to further optimize the accuracy–speed trade-off.
>
> ---
>
> We hope our responses have addressed the reviewer's concerns. If anything remains unclear or if you have further questions, we'd be happy to clarify and discuss them in more detail.
>
> ---
>
> [A]: *Yijun Yuan, et al. "SLAM-Former: Putting SLAM into One Transformer." arXiv preprint arXiv:2509.16909, 2025.*
> [B]: *Xintao Wang, et al. "Real-ESRGAN: Training real-world blind super-resolution with pure synthetic data." Proceedings of the International Conference on Computer Vision Workshops, 2021.*
> [C]: *Pin-Lun Hsu, et al. "Liger-Kernel: Efficient Triton Kernels for LLM Training." Championing Open-source DEvelopment in ML Workshop at ICML, 2025.*
> [D]: *Maksym Andriushchenko, et al. "Why Do We Need Weight Decay in Modern Deep Learning?" Advances in Neural Information Processing Systems, 2024.*
> [E]: *Joshua Ainslie, et al. "GQA: Training Generalized Multi-Query Transformer Models from Multi-Head Checkpoints." Proceedings of the Conference on Empirical Methods in Natural Language Processing, 2023.*
> [F]: *DeepSeek-AI, et al. "DeepSeek-V2: A Strong, Economical, and Efficient Mixture-of-Experts Language Model." arXiv preprint arXiv:2405.04434, 2024.*

---

### Official Review · Reviewer_P6AK · 2025-11-01

**Soundness:** 3
**Presentation:** 3
**Contribution:** 2
**Rating:** 4
**Confidence:** 5

**Summary:**

The paper tackles sequential, feed-forward 3D reconstruction, whereas most existing feed-forward models require processing all images at once. The core idea is a causal Transformer in which intra-frame self-attention is followed by inter-frame cross-attention to cached tokens from past frames, enabling online processing with KV-cache efficiency. A windowed streaming mode retains the first frame (to preserve a canonical world frame) plus the most recent context, bounding memory while maintaining global consistency. The model outputs local/global pointmaps with confidences and camera pose, and the authors fine-tune DUSt3R and VGGT backbones on public datasets. Experiments span video depth estimation and 3D reconstruction, comparing against feed-forward baselines and their pose-graph/global-alignment (GA) extensions. Results show that the proposed method delivers higher accuracy, markedly higher throughput, and linear (or constant, in windowed mode) memory in streaming/online settings without test-time global alignment, with gains especially clear on long sequences and under tight memory constraints.

**Strengths:**

### Originality

* To the best of my knowledge, this is among the first works to extend feed-forward 3D reconstruction to sequential processing via a causal transformer, rather than relying on pose-graph/global alignment.
* While it could use more elaboration, the [reg] token is an interesting mechanism to make the model explicitly aware of the anchor frame. In addition, omitting view embeddings is a novel choice that encourages order-agnostic generalization to different input image orders.

### Quality

* The comparisons against both feed-forward baselines and their global-alignment (GA) extensions are generally comprehensive, covering video depth and 3D reconstruction on short and long sequences.
* The generality of the approach is validated by combining it with both DUSt3R and VGGT baselines.

### Clarity

* The paper is well organized and motivated. The method is clearly introduced, and the evaluations are easy to follow.

### Significance

* The solution enables sequential reconstruction without pose-graph alignment, making it potentially complementary to pose-graph methods under conditions such as small overlaps.
* It imports ideas from LLMs into feed-forward 3D reconstruction, which can inspire follow-up research that leverages recent advances beyond just causal transformers.
* The method is general and composable with other transformer-based feed-forward models, and its efficiency could enable deployment in compute-limited applications such as robotics.

**Weaknesses:**

- Contribution may be limited: The gains appear to stem primarily from a causal transformer framework rather than components specific to 3D reconstruction. The paper should clarify what is fundamentally new beyond causal masking and standard transformer design, and what is uniquely tailored to 3D reconstruction.

- Anchor design needs more elaboration: The proposed [reg] token is interesting, but there is no controlled comparison to simpler anchors such as a global CLS token or relative view positional embeddings. This makes it hard to judge when [reg] is necessary and sufficient. The paper also lacks experiments on unordered image collections that could demonstrate the hypothesized advantage.

- Experimental scale and fairness: Most experiments rely on small-scale data (e.g., 7-scenes), which weakens claims—especially against pose-graph alignment methods. 7-scenes has strong inter-image connectivity and may not expose long-sequence drift. Since attention couples camera poses and depths, it is unclear whether bundle-adjustment refinement is applied, which affects fairness.

- Missing robustness and failure analysis: The paper offers limited analysis of failure cases to clarify pros/cons relative to global alignment approaches.

**Questions:**

- Please refere to the weakness
- What happens if the canonical first frame is occluded or does not have overlap with latest frames? Can the model re-anchor and attentive to other frames?
- How the performance is compared with VGGT-SLAM, which is a pose graph alignment based method built upon VGGT.

---

> ### Author Response · Authors · 2025-11-25
> **Rebuttal by Authors (Part I)**
>
> We thank the reviewer for acknowledging the comprehensive nature of our experiments and the significance of our approach. We also appreciate the reviewer's constructive feedback and address specific concerns in detail below.
>
> ---
>
> ### W1: Clarification on Contributions
>
> Our proposed method is specifically designed for streaming 3D reconstruction, leveraging advances in Large Language Models (LLMs) architectures such as KV caching and causal masking. This problem has been largely underexplored by existing approaches (e.g., VGGT), which rely on full-attention architectures and primarily target multi-view reconstruction.
>
> Instead of introducing unnecessary architectural complexity, we aim to develop a clean and unified design consistent with the decoder-only transformer paradigm widely adopted in modern LLMs. This choice enhances the generality and scalability of our method, allowing it to naturally benefit from ongoing progress in the LLM field.
>
> For example, our variant STream3R-W, which integrates window attention into the streaming 3D reconstruction pipeline, achieves the highest FPS among compared methods (see Tab. 2 of the main paper) without sacrificing, and even improving performance on video depth estimation (Bonn dataset). This highlights an important direction for future research on efficient streaming 3D perception. Moreover, adopting window-based attention naturally reduces the number of processed tokens, which facilitates online downstream applications such as visual SLAM, VLA, and 3D tracking. Besides, our ablation in Sec. 5.3 also confirms that our model converges faster than CUT3R with 60% training speedup, showing the effectiveness of our decoder-only transformer against the existing streaming 3D reconstruction method CUT3R's RNN design.
>
> ---
>
> ### W2: Anchoring / [reg] Token
>
> Thanks for your question. We would like to clarify that the [reg] token is common and implemented in a similar way as Fast3R and VGGT, which is a standard strategy and not the main contribution of our proposed framework.
>
> Since our method is designed to work on both unordered inputs (multi-view images) and ordered inputs (streaming videos), relative view positional embeddings (i.e., the PE used in video diffusion models) cannot be used here. Some 3D view-relevant positional embedding designs like ProPE [A] require posed inputs, whereas our method uses RGB-only inputs, so it cannot be adopted either.
>
> Our training data includes both video sequences and unordered photo collections. As shown in Tab. 3 and Tab. 7, our 3D reconstruction experiments use the sparse multi-view images from 7-Scenes and NRGBD, which effectively serve as unordered image inputs. Under this setting, our method achieves state-of-the-art performance for streaming 3D reconstruction. We further add visualizations for unordered image inputs and even the case with the non-overlapping anchoring view in Fig. 5 of the appendix.

---

> ### Author Response · Authors · 2025-11-25
> **Rebuttal by Authors (Part II)**
>
> ### W3: Experimental Scale and Fairness
>
> We follow the CUT3R evaluation protocol for a fair comparison and have conducted extensive experiments across multiple datasets and tasks (Sec. 5). For example:
>
> - Video depth estimation on *Sintel*, *BONN*, and *KITTI* (with more than 100 frames per scene),
> - 3D reconstruction on *7-Scenes* and *NRGBD*,
> - Camera pose estimation on *Sintel*, *TUM-dynamics*, and *ScanNet*.
>
> To further verify performance on large-scale data with longer sequences, we include 3D reconstruction experiments on ETH3D, as shown in the Table below.
>
> ***Table A: 3D Reconstruction Comparison on ETH3D.***
> | Method | Type | Acc ↓ (Mean / Med.) | Comp ↓ (Mean / Med.) | NC ↑ (Mean / Med.) |
> | :--- | :---: | :---: | :---: | :---: |
> | DUSt3R-GA | Optim. | 2.582 / 2.034 | 2.126 / 1.544 | 0.548 / 0.573 |
> | MASt3R-GA | Optim. | 2.682 / 2.458 | 2.206 / 1.734 | 0.531 / 0.540 |
> | | | | | | | |
> | Fast3R | FA | 0.832 / 0.691 | 0.978 / 0.683 | 0.667 / 0.766 |
> | VGGT | FA | **0.280** / **0.185** | 0.305 / 0.182 | **0.853** / **0.950** |
> | | | | | | | |
> | CUT3R | Stream. | 0.617 / 0.525 | 0.747 / 0.579 | 0.754 / 0.848 |
> | Span3R | Stream. | 1.730 / 1.107 | 1.373 / 0.742 | 0.545 / 0.634 |
> | SLAM3R | Stream. | 1.678 / 1.288 | 0.996 / 0.499 | 0.615 / 0.681 |
> | **Ours** | Stream. | **0.363** / **0.227** | **0.245** / **0.094** | **0.812** / **0.943** |
> | | | | | | | |
>
> As can be seen, GA (global alignment)-based methods (DUSt3R, MASt3R) performs significantly worse than feed-forward reconstruction methods (CUT3R and Ours), indicating that they fail to generalize to challenging scene and long video sequences. Besides, our method significantly outperforms other streaming approaches (CUT3R, Span3R, SLAM3R). While the full-attention offline method VGGT performs great, our streaming method achieves the best Comp score among all methods (0.245 vs VGGT 0.305) and remains competitive in accuracy.
>
> For all results in the paper, we do **not** apply any post-processing optimization, such as bundle-adjustment refinement. However, the post-processing optimization is orthogonal to our contribution and can be included to further improve the performance.
>
> ---
>
> ### W4: Robustness Analysis and Global Alignment-based methods
>
> Early research along this line of work like Dust3R and Monst3R relies on global alignment to register multi-view (>2) images input. We have discussed in the paper (Tab. 1, 2, 3) that our proposed method consistently outperforms this line of work by a clear margin. In Fig. 3 and the uploaded demo video, we qualitatively show that our proposed method yields better performance.
>
> ---
>
> ### Q2: First Frame Robustness
>
> Thanks for the question. Using the first frame as the global coordinate system is a standard convention across DUSt3R and its follow-up works, including MASt3R, MonST3R, CUT3R, VGGT, and ours. We would like to highlight that the 3D reconstruction experiments reported in Tab. 3 and Tab. 7 of the main paper are evaluated on sparse or even non-overlapping inputs, and the results already demonstrate that our method remains robust even when the first frame has limited overlap. Furthermore, as shown in the newly added Fig. 5(b) in the appendix, even when the first frame has very little overlap, our model still shows strong implicit relative pose-learning capability for the other views.
>
> Quantitatively, we further follow the degradation pipeline of Real-ESRGAN [B] to corrupt the first frame of each sequence, and then evaluate VGGT, CUT3R, and our method on the 7-Scenes dataset. This directly examines the scenario mentioned by the reviewer (e.g., the first frame being occluded). As shown in Tab. B, all methods experience some degradation. However, CUT3R's Acc error increases significantly from 0.126 to 0.335, whereas STream3R only increases from 0.122 to 0.223, demonstrating that our method is considerably more robust under such challenging conditions.
>
> ***Table B: Impact of First-View Degradation on 3D Reconstruction (7-Scenes).***
>
> | Method | Acc (Mean) ↓ | Acc (Med.) ↓ | Comp (Mean) ↓ | Comp (Med.) ↓ | NC (Mean) ↑ | NC (Med.) ↑ |
> | :--- | :---: | :---: | :---: | :---: | :---: | :---: |
> | **VGGT** | 0.087 | 0.039 | 0.091 | 0.039 | 0.787 | 0.890 |
> | **CUT3R** | 0.126 | 0.047 | 0.154 | 0.031 | 0.727 | 0.834 |
> | **Ours** | 0.122 | 0.044 | 0.101 | 0.038 | 0.746 | 0.856 |
> | **VGGT** (w/ 1st view deg.) | 0.144 (+0.057) | 0.062 (+0.023) | 0.172 (+0.081) | 0.060 (+0.021) | 0.708 (-0.079) | 0.811 (-0.079) |
> | **CUT3R** (w/ 1st view deg.) | 0.335 (+0.209) | 0.270 (+0.223) | 0.320 (+0.166) | 0.276 (+0.245) | 0.666 (-0.061) | 0.752 (-0.082) |
> | **Ours** (w/ 1st view deg.) | 0.223 (+0.101) | 0.117 (+0.073) | 0.214 (+0.113) | 0.139 (+0.101) | 0.695 (-0.051) | 0.789 (-0.067) |

---

> ### Author Response · Authors · 2025-11-25
> **Rebuttal by Authors (Part III)**
>
> ### Q3: Comparison with VGGT-SLAM
>
> Following the reviewer's suggestion, we compare our method with VGGT-SLAM [C] on both static scenes (NRGBD) and dynamic scenes (Sintel and TUM-dynamics). As shown in Tab. C, our approach performs on par with SLAM-specialized techniques on static scene reconstruction. Moreover, as demonstrated in Tab. D below and the camera trajectory visualization in Appendix Fig. 6, our method is also able to reconstruct dynamic scenes, a capability that conventional SLAM-based methods typically lack.
>
> We also emphasize that our method targets a different problem setting from SLAM-based approaches. Our goal is to develop a unified streaming 3D/4D reconstruction pipeline capable of handling both foreground and background regions, whereas SLAM-based methods primarily focus on reconstructing static backgrounds and estimating accurate camera poses.
>
> Despite these differing objectives, our approach is fully compatible with feed-forward SLAM systems and can be seamlessly integrated into their pipelines. As demonstrated in a recent work SLAM-Former [D], streaming-based 3D reconstruction with KV caching can effectively support frontend tasks such as keyframe selection, tracking, and mapping within a SLAM system.
>
> ***Table C: 3D Reconstruction Comparison on Dense NRGBD (~150 frames)***
>
> | Method | Acc (Mean) ↓ | Acc (Med.) ↓ | Comp (Mean) ↓ | Comp (Med.) ↓ | NC (Mean) ↑ | NC (Med.) ↑ |
> | :--- | :---: | :---: | :---: | :---: | :---: | :---: |
> | **VGGT-SLAM** | **0.039** | **0.017** | 0.028 | 0.009 | **0.781** | **0.939** |
> | **Ours** | 0.046 | 0.02 | **0.012** | **0.005** | 0.756 | 0.923 |
>
> ***Table D: Camera Pose Comparison with VGGT-SLAM***
>
> | Method | Sintel (ATE) ↓ | Sintel (RPE trans) ↓ | Sintel (RPE rot) ↓ | TUM-dynamics (ATE) ↓ | TUM-dynamics (RPE trans) ↓ | TUM-dynamics(RPE rot) ↓ |
> | :--- | :---: | :---: | :---: | :---: | :---: | :---: |
> | **VGGT-SLAM** | 0.305 | 0.082 | 4.14 | 0.041 | 0.014 | 0.879 |
> | **Ours** | **0.213** | **0.076** | **0.868** | **0.026** | **0.013** | **0.330** |
>
> ---
>
> We hope our responses have addressed the reviewer's concerns. If anything remains unclear or if you have further questions, we'd be happy to clarify and discuss them in more detail.
>
> ---
>
> [A]: *Ruilong Li, et al. "Cameras as Relative Positional Encoding." Advances in Neural Information Processing Systems, 2025.*
> [B]: *Xintao Wang, et al. "Real-ESRGAN: Training real-world blind super-resolution with pure synthetic data." Proceedings of the International Conference on Computer Vision Workshops, 2021.*
> [C]: *Dominic Maggio, et al. "VGGT-SLAM: Dense RGB SLAM Optimized on the SL (4) Manifold." Advances in Neural Information Processing Systems, 2025.*
> [D]: *Yijun Yuan, et al. "SLAM-Former: Putting SLAM into One Transformer." arXiv preprint arXiv:2509.16909, 2025.*

---

### Author Response · Authors · 2025-11-26
**General Response by Authors**

We thank the reviewers for their constructive feedback on our streaming 3D reconstruction framework. We are glad that the reviewers appreciate the novelty and originality of our causal transformer paradigm (```P6AK```, ```JPV9```, ```vTS9```). The reviewers highlighted our method's scalability and efficiency (```P6AK```, ```JPV9```) alongside its strong performance compared to existing methods (```JPV9```, ```vTS9```). Reviewers also mentioned that our windowed variant is particularly noteworthy (```JPV9```) and has great potential for downstream tasks in compute-limited scenarios (```P6AK```). Finally, we are encouraged by the recognition of our comprehensive experiments (```JPV9```, ```shrj```, ```P6AK```) and the clarity of our presentation and demos (```P6AK```, ```shrj```).

We carefully reviewed the reviewers’ feedback and added extensive experiments and analyses, including 7 additional analysis sections incorporated into the Appendix of the revised manuscript. A summary of the major updates is provided below:

- **Long-Sequence Evaluation:** To address concerns regarding drift and error accumulation, we added experiments on the NRGBD and 7-Scenes datasets with varying frame intervals (Tab. 15). We also evaluated extremely long sequences with 1.5k frames (Tab. 16).

- **Windowed Variant Analysis:** We provided a detailed analysis of window sizes for STream3R-W to clarify the trade-off between memory usage and accuracy (Tab. 14).

- **Robustness Study:** We analyzed robustness against first-frame degradation (Tab. 10). We also included qualitative results for edge cases where the anchoring view does not overlap with subsequent views (Fig. 5).

- **Expanded Comparisons:** We included complete comparisons with the SLAM-based method VGGT-SLAM (Tab. 12, 13, Fig. 6) and added 3D reconstruction evaluation on the ETH3D dataset (Tab. 9).

We will release the code and pretrained models to support reproducibility and future research in the community. For detailed responses to each reviewer, please find them below.

---

> ### Author Response · Authors · 2025-12-01
> **Follow-Up Response to the AC and Reviewers**
>
> We thank the AC and all reviewers for their thoughtful engagement with our work, especially during the challenging period of the recent OpenReview incident.
>
> Although the review window has closed, we would like to emphasize that we have provided all the discussions and experimental results requested by the reviewers. We would particularly like to highlight the positive feedback on our work, especially from reviewer vTS9, who offered strong acknowledgment. In addition, we have provided the detailed discussions and comparisons requested by reviewer shrJ, who indicated that they would consider raising their score. Please let us know if any further discussions or experiments are required to resolve the remaining concerns. We thank the AC and all the reviewers for their insights on this work, and we truly appreciate the AC’s additional commitment during this difficult time.

---

### Public Comment · ~Héctor_Carrión1 · 2026-03-03

Congratulations to the authors for their acceptance to the conference! I am a big fan of Stream3r and the authors have been extremely engaged with the community in supporting their open source work on GitHub.

---

### Meta-Review · Area_Chair_57Q8 · 2025-12-31

**Summary:**

The paper received mixed initial reviews, with scores of 8, 6, 4, and 4. Reviewers generally agreed that the paper is technically sound, well written, and presents a compelling approach to scalable streaming 3D reconstruction. A commonly recognized strength is the architectural design that adapts a standard LLM-style causal Transformer with KV caching to the streaming 3D reconstruction setting, supported by extensive empirical validation across depth, reconstruction, and pose benchmarks. At the same time, several reviewers raised concerns regarding the degree of conceptual novelty, the reliance on first-frame anchoring, robustness and potential drift over long sequences, and comparisons to closely related methods such as StreamVGGT and VGGT-SLAM.

In the rebuttal and revised version, the authors provided a thorough and responsive set of additional experiments and clarifications. These include extended long-sequence evaluations on NRGBD and 7-Scenes, robustness tests with degraded first frames, new ETH3D results, explicit comparisons to VGGT-SLAM, and more detailed analyses of the windowed variant and its memory–quality trade-offs. Multiple reviewers’ technical concerns were responded to through these additions. While some conceptual skepticism—particularly around novelty and framing—may persist for one reviewer, the AC anticipates that the final score distribution is likely to move toward 8, 8, 6, 6 or 8, 6, 6, 6, with a remaining possibility of 8, 6, 6, 4 (see detailed discussion in Reviewer Concerns and Reviewer Scores).

From the AC’s perspective, this submission makes a meaningful and valid contribution by demonstrating that a standard LLM-like causal Transformer architecture, when carefully adapted, can be effective for streaming and scalable 3D reconstruction, leading to clear improvements in reconstruction quality over prior approaches. While the conceptual ingredients draw on well-established ideas from sequence modeling, the paper convincingly shows that this architectural paradigm transfers effectively to dense 3D regression, offering clear scalability and efficiency benefits. The AC believes this work can inspire the 3D community to further explore modern large-model infrastructures and architectural design paradigms. Taking the technical quality, experimental thoroughness, and relevance to the community together, the AC recommends acceptance of this paper, viewing it as a solid step toward bridging modern sequence-modeling architectures and practical large-scale 3D reconstruction.

**Reviewer Concerns:**

### Reviewer P6AK (Score: 4)

- The reviewer raised concerns that the main contribution may be limited, arguing that the gains largely stem from applying a generic causal Transformer framework rather than introducing 3D-specific innovations. Additional concerns included insufficient analysis of the first-frame anchoring mechanism (the [reg] token), limited robustness and failure-case analysis and the need for stronger evaluation on longer sequences beyond standard benchmarks such as 7-Scenes. The reviewer also requested explicit comparison to VGGT-SLAM and clearer discussion of the role and impact of the first frame.
- In the rebuttal, the authors clarified the novelty as reframing streaming 3D reconstruction into a decoder-only causal Transformer with KV caching inspired by LLM inference. They added new ETH3D results, robustness tests with degraded first frames, and explicit comparisons against VGGT-SLAM on both static and dynamic scenes. The anchoring design was also clarified.

---

### Reviewer JPV9 (Score: 6)

- The reviewer expressed concerns about potential error accumulation and drift over long sequences, reliance on first-frame anchoring without sufficient robustness analysis, and whether the claim of “LLM-style training infrastructure” was sufficiently supported, given that most evidence focused on inference-time efficiency.
- In response, the authors added long-sequence evaluations on NRGBD with varying frame intervals and extended sequence lengths, included experiments degrading the first frame to test anchoring robustness, and clarified the meaning of “LLM-style training” by detailing adopted kernel optimizations and architectural choices, while acknowledging limitations and future extensions.

---

### Reviewer vTS9 (Score: 8)

- The reviewer raised questions about how STream3R differs from concurrent StreamVGGT and why it performs better, and noted that some highlighted claims (e.g., world vs. local pointmaps, LLM-style training, and novel view synthesis generalization) were not sufficiently supported by experiments in the original submission.
- In the rebuttal, the authors clarified the key differences relative to StreamVGGT, including memory scaling behavior, training strategy, and metric-scale reconstruction. They added discussion and experiments on local versus global pointmaps and clarified the scope of claims related to LLM-style training and downstream usage.

---

### Reviewer shrJ (Score: 4)

- The reviewer raised concerns about strong architectural similarity to VGGT, questioned whether causal attention alone justifies the contribution, and noted that the windowed variant (STream3R-W) was not consistently evaluated across tasks. The reviewer also pointed out that the original reconstruction experiments were too short to meaningfully assess streaming behavior and requested ETH3D results and longer-sequence evaluations.
- In the rebuttal, the authors added extensive long-sequence experiments on NRGBD and 7-Scenes across different frame intervals, provided detailed window-size ablations illustrating quality–memory trade-offs, added ETH3D reconstruction results, and clarified the practical role of STream3R-W under fixed memory constraints.

---
Overall, the AC finds that the authors provided a thorough and responsive rebuttal, responding to nearly all of the concerns raised by the reviewers through additional experiments, analyses, and clarifications.

**Reviewer Scores:**

### Reviewer P6AK

- **Original score:** 4
- **Predicted final score:** 4–6
- **Rationale:** The rebuttal addresses the reviewer’s primary technical concerns by adding longer-sequence evaluations, robustness tests for first-frame anchoring, and explicit comparisons to VGGT-SLAM. While the requested empirical evidence is largely provided, the AC notes that the reviewer may have a different taste from the paper’s overall direction, and some skepticism regarding conceptual novelty may remain. As a result, a score increase is possible but not guaranteed.

---

### Reviewer JPV9

- **Original score:** 6
- **Predicted final score:** 6–8
- **Rationale:** The rebuttal strengthens the empirical validation of scalability, long-sequence robustness, and anchoring stability, directly engaging with the reviewer’s concerns. Since the reviewer was already mildly positive and most issues are addressed, the score is likely to remain positive or potentially increase.

---

### Reviewer vTS9

- **Original score:** 8
- **Predicted final score:** 8
- **Rationale:** The reviewer was already strongly positive, and the rebuttal addresses the clarificatory questions regarding comparisons to StreamVGGT and the scope of the claims. The reviewer also explicitly mentioned that their rating would remain unchanged.

---

### Reviewer shrJ

- **Original score:** 4
- **Predicted final score:** 6
- **Rationale:** The rebuttal responds to the major concerns raised by this reviewer, including long-sequence evaluation, ETH3D results, and a more systematic analysis of the windowed variant. Given that the reviewer stated a willingness to raise their score if these issues were addressed, a score increase is likely.

---

### Decision · Program_Chairs · 2026-01-26

Accept (Poster)